# Can Information-Theoretic Generalization Bound Explain the Generalization of Pre-trained Language Model?

## Abstract

Although language models exhibit exceptional generalization capabilities in downstream tasks after extensive text pre-training, the underlying causes behind this generalization remain unclear. Existing studies on information-theoretic generalization bounds suggest that the compression of **i**nformation stored **i**n the **w**eights (IIW) is a crucial factor influencing a model's ability to generalize, with some experiments indicating a correlation between lower IIW and improved generalization. However, it remains uncertain whether IIW is applicable to pre-trained language models. In this work, we find that using IIW can explain why the pre-trained language models have better generalization compared to non-pre-trained language models. Unfortunately, we also discover that IIW does not consistently reflect the degree of generalization when applying IIW to study the fine-tuning process of pre-trained language models. We revisit existing IIW estimation methods, highlighting their limitations in accurately estimating IIW based on theoretical and empirical evidence. Our findings suggest that current information-theoretic generalization bounds, constrained by the limitations of IIW estimation methodologies, fail to accurately capture the generalisation performance of pre-trained language models.

## 1 Introduction

Natural Language Processing (NLP) has undergone a significant transformation with the advent of large-scale pre-trained language models (Radford et al., 2018; Devlin et al., 2018; Liu et al., 2019; Clark et al., 2020; Touvron et al., 2023a;b; Bai et al., 2023). These models, predominantly based on the Transformer architecture (Vaswani et al., 2017), utilize either the encoder or decoder structure. Through extensive pre-training on vast amounts of raw text data and subsequent fine-tuning for specific downstream tasks, these models demonstrate remarkable generalization capabilities across a diverse array of applications, including sentiment analysis, machine translation, and question answering. While the Transformer's encoder is generally employed for natural language understanding and the decoder for natural language generation, both structures adhere to the same training paradigm: pre-training then fine-tuning. Despite the notable success of this approach, the underlying reasons for the effectiveness of pre-training then fine-tuning paradigm, especially in comparison to random initialization then fine-tuning on downstream tasks, remain not fully understood.

Recent advancements in information theory offer new insights into model generalization (Xu & Raginsky, 2017; Russo & Zou, 2016; Negrea et al., 2019; Wang & Mao, 2023). Specifically, information-theoretic generalization bounds suggest that the compression of **information stored in weights** (IIW) is a key factor influencing a model's ability to generalize. A lower IIW means a smaller upper bound on generalization error, which typically means that the model has better generalization performance(Xu & Raginsky, 2017). Building on these insights, (Wang et al., 2022) derive an approximation of tractable IIW and establish an IIW-based information bottleneck, focusing on the trade-off between model generalization and information complexity. They empirically observe a two-phase learning process, consisting of an initial learning phase followed by a forgetting phase, with models that exhibit lower IIW after this process tending to generalize better. In a related study, (Song et al., 2024) apply IIW to measure model generalization in video object segmentation tasks, finding that models with superior generalization performance consistently have lower IIW.

One limitation of the work by (Wang et al., 2022) is that the effectiveness of IIW has only been validated on multi-layer perceptrons (MLP) and convolutional neural networks (CNN). It is currently unknown whether IIW can also explain the generalization capabilities of pre-trained language models. Although pre-trained language models have achieved good performance on downstream tasks, their effectiveness remains unknown. Therefore, it is crucial to investigate whether IIW can also explain the generalization capabilities of pre-trained language models.

In this work, we extend the application of information-theoretic generalization bounds to pre-trained language models. Specifically, we first compare the IIW (Information in Weights) of pre-trained and non-pre-trained language models on downstream tasks. Our findings indicate that pre-trained language models exhibit lower IIW compared to non-pre-trained language models, suggesting they have a lower generalization error upper bound and better performance in downstream tasks. We also observe differences in layer-specific IIW between pre-trained and non-pre-trained language models. For non-pre-trained language models, the fc and emedding layers have the highest IIW, and in some cases the IIW of the embedding layer exceeds that of the fc layer. In pre-trained language models, the fully connected (fc) layer has significantly higher IIW than other layers and the IIW of the embedding layer will be significantly lower than that of the fc layer. When we attempt to use IIW as a generalization proxy during the fine-tuning of pre-trained language models, we find that lower IIW does not always correlate with better generalization performance. Even within the same pre-trained language model fine-tuned using different methods, a lower IIW does not consistently indicate improved generalization. This prompts us to reconsider whether existing information-theoretic generalization bounds can reliably serve as generalization proxies. We believe the reason for these experimental outcomes lies in the current IIW computation methods, which may lack precision. We provide both theoretical analysis and empirical evidence to support this claim, suggesting that more refined IIW computation techniques are needed for it to effectively serve as a generalization error proxy. The contributions of this work can be summarized as follows:

- We compare the IIW of pre-trained and non-pre-trained language models across different datasets and find that pre-trained language models consistently exhibit lower IIW and better performance, indicating that IIW can be used to explain the superiority of pre-trained language models compared to non-pre-trained language models. Additionally, the IIW of the fully connected (fc) layer in pre-trained language models is significantly higher than in other layers.
- We observe that IIW cannot reliably serve as a generalization proxy during the fine-tuning of pre-trained language models. Even when fine-tuning the same pre-trained language model using different methods, lower IIW does not necessarily correspond to better generalization performance.
- We find that the existing method for calculating IIW prevents IIW from serving as a generalization proxy for pre-trained language models during fine-tuning. We theoretically prove that there is indeed some degree of error in the precision estimation of the current IIW. We also provide empirical evidence to support this theoretical analysis.

## 2 PRELIMINARY

### 2.1 NOTATION

Let $\mathcal{Z}$ be an instance space, $\mathcal{W}$ be a hypothesis space and $\ell : \mathcal{W} \times \mathcal{Z} \to \mathbb{R}^+$ is a nonnegative loss function. We denote a training set with $n$ samples as $Z = (Z_1, \cdots, Z_n) \in \mathcal{Z}^n$. Typically, we assume that the data satisfies the independent and identically distributed (i.i.d.) assumption. Furthermore, we define the distribution of the training set $Z$ as $P_Z$. Based on the training set, the learning algorithm selects the hypothesis $W$ from the hypothesis space $\mathcal{W}$. The learning algorithm $P_{W|Z}$ is defined as a probabilistic mapping from the training set $Z$ to the hypothesis $W$. The population risk of a hypothesis $w \in \mathcal{W}$ on $P_Z$ is

$$L_{P_Z}(w) \triangleq \mathbb{E}_{P_Z}[\ell(w, Z)] = \int_{\mathcal{Z}} \ell(w, z)p(z)dz \tag{1}$$

The goal of learning is to minimize the population risk of the output hypothesis under the data generating distribution $P_Z$. In actual training, since $P_Z$ is unknown, it is impossible to directly calculate the population risk $L_{P_Z}(w)$ for any $w \in \mathcal{W}$. Therefore, we can only use a finite training set

$Z$ to calculate the training loss of hypothesis $w$ as a proxy, defined as

$$L_Z(w) \triangleq \frac{1}{n} \sum_{i=1}^{n} \ell(w, Z_i) \tag{2}$$

The training loss is also known as the empirical risk. By utilizing the computable training loss, the goal of learning is to minimize the training loss of the output hypothesis while ensuring that the difference between the population risk and the training loss is small. This is measured by the generalization error

$$\text{gen}(w, Z) \triangleq L_{P_Z}(w) - L_Z(w) \tag{3}$$

which is also called the generalization gap.

## 2.2 INFORMATION-THEORETIC GENERALIZATION BOUND

(Xu & Raginsky, 2017) derived upper bound on the generalization error of a learning algorithm in terms of the mutual information between training set and hypothesis $I(W; Z)$. It means that the compression of information stored in weights (IIW), i.e. $I(W; Z)$, is proved to play a key role in generalization.

**Theorem 2.1** ((Xu & Raginsky, 2017)). *Suppose $\ell(w, Z)$ is $\sigma$-subgaussian under $P_Z$ for all $w \in \mathcal{W}$, then*

$$|\mathbb{E}_{P_{WZ}}[gen(W, Z)]| = |\mathbb{E}_{P_{WZ}}[L_{P_Z}(W) - L_Z(W)]| \leq \sqrt{\frac{2\sigma^2}{n} I(W; Z)}$$

## 2.3 ESTIMATION OF IIW

Although (Xu & Raginsky, 2017) proposed the information-theoretic generalization bound, (Xu & Raginsky, 2017) did not provide a solution for IIW. To address this issue, (Wang et al., 2022) proposed an algorithm for the efficient approximation of IIW. Specifically, according to the definition of mutual information, for $I(W; Z)$, we have:

$$I(W; Z) = \mathbb{E}_{P_Z}[\text{KL}(p(W|Z) \parallel p(W))] \tag{4}$$

To get the closed-form solution of Kullback-Leibler (KL) divergence term in Eq.(4), (Wang et al., 2022) assume both $p(W) = \mathcal{N}(W \mid \theta_0, \Sigma_0)$ and $p(W|Z) = \mathcal{N}(W \mid \theta_Z, \Sigma_Z)$ are Gaussian distributions, then KL term can be expressed as

$$\text{KL}(p(W|Z \parallel p(W)) = \frac{1}{2}[\log \frac{\det \Sigma_Z}{\det \Sigma_0} - D + (\theta_Z - \theta_0)^T \Sigma_0^{-1}(\theta_Z - \theta_0) + \text{tr}(\Sigma_0^{-1}\Sigma_Z)] \tag{5}$$

where det A and tr(A) are the determinant and trace of matrix A, respectively; D is the dimension of $W$. To simplify the computation, (Wang et al., 2022) further assume that the covariances of the prior and posterior are proportional. Consequently, the logarithmic and trace terms in Eq. (5) both become constant. The $I(W; Z)$ can be express as

$$I(W; Z) \propto \mathbb{E}_{P_Z}[(\theta_Z - \theta_0)^T \Sigma_0^{-1}(\theta_Z - \theta_0)] \tag{6}$$

where $\theta_Z$. By using influence functions (Ling, 1984) and Poisson bootstrapping, (Wang et al., 2022) approximated the covariance matrix of the oracle prior. As a result, $I(W; Z)$ can be rewritten as:

$$I(W; Z) \propto n\mathbb{E}_{P_Z}[(\theta_Z - \theta_0)^T F_\theta(\theta_Z - \theta_0)] \tag{7}$$

where $F_\theta$ is Fisher information matrix (FIM).

## 3 EXPERIMENT

In this section, we design experiments to evaluate whether IIW can be used to explain the generalization capabilities of pre-trained language models. Specifically, in Sec3.2, we aim to verify if IIW can account for the performance differences between pre-trained and non-pre-trained language models on downstream tasks. In Sec3.3, we explore whether IIW can serve as a generalization proxy during the fine-tuning process of pre-trained language models. Finally, in Sec3.4, we investigate whether IIW can be used to assess the generalization performance of models obtained from the same pre-trained language model but fine-tuned using different methods.

## 3.1 SETUP

In this experiment, we evaluate the performance of three pre-trained language models: BERT-base (Devlin et al., 2018), RoBERTa-base (Liu et al., 2019) and ELECTRA-base (Clark et al., 2020). The evaluation is conducted on several NLP datasets (Wang, 2018), including Recognizing Textual Entailment (RTE), Question Natural Language Inference (QNLI), Multi-Genre Natural Language Inference (MNLI), MNLI-Mismatch, Quora Question Pairs (QQP), Microsoft Research Paraphrase Corpus (MRPC), and Stanford Sentiment Treebank 2 (SST-2).

Due to these datasets do not provide public test set labels, we use the development set as the test set for reporting results, consistent with previous research methodologies (Wu et al., 2020). The training set for each dataset is randomly split into a 9:1 ratio to create the train and development sets. For all models, we set the seed to 42 to ensure reproducibility. The experiments are conducted with a batch size of 32 and run for 5 epochs. We explore two learning rates: 4e-5 and 3e-5. The learning rate schedule includes using the Adam optimizer (Kingma & Ba, 2014) with a warm-up proportion of 0.1.

**Evaluation metric.** To calculate the information in the weights (IIW), we utilize the open-source code provided by (Wang et al., 2022).

## 3.2 IIW OF PRE-TRAINED LANGUAGE MODEL AND NON-PRE-TRAINED LANGUAGE MODEL

**Pre-training results.** The effectiveness of pre-trained language models is demonstrated not only by their superior performance across various downstream tasks but also by their significantly better results compared to non-pre-trained language models, a fact that has been consistently validated in numerous empirical studies. From the perspective of information-theoretic generalization bounds, if a pre-trained language model exhibits better generalization, it implies that the upper bound on its generalization error is lower, meaning the IIW is smaller. Specifically, we compared the IIW of pre-trained language models with randomly initialized models across different datasets. As shown in Table1, it is evident that pre-trained language models consistently outperform in all test sets and have correspondingly lower IIW. A lower IIW indicates that the pre-trained language models memorize less information from the dataset. This result aligns with intuition since pre-trained language models have already undergone self-supervised learning on large amounts of raw text and have acquired semantic knowledge of the text. For example, BERT can complete sentences with masked words, requiring less task-specific information to perform well.

**Results for different layers.** Previous research suggests that different layers of pre-trained language models play distinct roles (Rogers et al., 2021). To explore this further, we compared the IIW differences between pre-trained and non-pre-trained language models at the layer level. Fig1 illustrates the IIW across various layers for both pre-trained and non-pre-trained language models. Detailed IIW value for each layer of the pre-trained language models can be found in the Appendix B. A consistent observation from Fig1 is that, in pre-trained language models, the IIW of the fully connected (fc) layer is significantly higher than that of other layers. We believe this is because the fc layer is added specifically to adapt to downstream tasks and is therefore not included in the pre-training process; instead, it is randomly initialized. This creates an "information gap" between the fc layer and the other layers, requiring the fc layer to "memorize" more information from the training set to compensate for this difference. This also highlights that the knowledge acquired during pre-training helps reduce the model's reliance on the training set. In non-pre-trained language models, although the IIW of the fc layer is relatively high, it is not the highest nor significantly greater than that of other layers. Comparing the IIW in the Embedding layer between pre-trained and non-pre-trained language models, Fig1 shows that the Embedding layer of non-pre-trained language models generally has a higher IIW, and in some cases, it is even the highest, such as in ELECTRA on SST-2. In contrast, the Embedding layer of pre-trained language models has a lower IIW. We believe this is because pre-trained language models learn more general token embeddings during the pre-training phase, which reduces their dependence on the dataset during fine-tuning. Another observation regarding the Embedding layer in pre-trained language models is that its IIW is typically higher than that of layer 0, with the exception of RoBERTa on MRPC. We believe this is due to the distinct functions of different layers. The Embedding layer is responsible for mapping raw tokens to token embeddings, while the subsequent layers manage the contextual interactions between the input tokens.

Table 1: The IIW and test accuracy (Acc) of pre-trained and non-pre-trained language models across various datasets. "Pretrain" indicates initialization with a pre-trained language model, while "Random" refers to random initialization. A green background denotes lower IIW, while a red background indicates higher IIW.

| Dataset | Metric | BERT | | RoBERTa | | ELECTRA | |
|---|---|---|---|---|---|---|---|
| | | Pretrain | Random | Pretrain | Random | Pretrain | Random |
| RTE | IIW | 2.54e-3 | 0.0185 | 5.37e-4 | 0.0059 | 2.03e-03 | 0.0186 |
| | Acc | 0.6751 | 0.5162 | 0.7581 | 0.4729 | 0.7870 | 0.4801 |
| MRPC | IIW | 1.50e-04 | 0.4143 | 1.53e-04 | 0.0145 | 5.31e-04 | 0.0204 |
| | Acc | 0.8162 | 0.6838 | 0.8922 | 0.6667 | 0.8873 | 0.6667 |
| MNLI-m | IIW | 7.10e-09 | 1.39e-06 | 4.00e-09 | 1.63e-06 | 4.65e-09 | 2.01e-06 |
| | Acc | 0.8358 | 0.6655 | 0.8620 | 0.6234 | 0.8744 | 0.6564 |
| MNLI-mm | IIW | 3.53e-09 | 1.39e-06 | 4.00e-09 | 1.63e-06 | 4.65e-09 | 2.02e-06 |
| | Acc | 0.8361 | 0.6694 | 0.8621 | 0.6276 | 0.8764 | 0.6585 |
| QNLI | IIW | 3.43e-09 | 3.44e-05 | 1.48e-08 | 3.03e-05 | 3.21e-08 | 2.06e-05 |
| | Acc | 0.9107 | 0.5843 | 0.9196 | 0.5779 | 0.9218 | 0.574 |
| QQP | IIW | 1.49e-09 | 1.87e-09 | 2.11e-09 | 3.60e-09 | 4.61e-09 | 7.97e-07 |
| | Acc | 0.9094 | 0.6982 | 0.9107 | 0.6318 | 0.9204 | 0.7997 |
| SST-2 | IIW | 9.60e-09 | 1.83e-06 | 6.92e-08 | 7.05e-06 | 7.37e-08 | 4.02e-06 |
| | Acc | 0.9197 | 0.7936 | 0.9346 | 0.8028 | 0.9392 | 0.8085 |

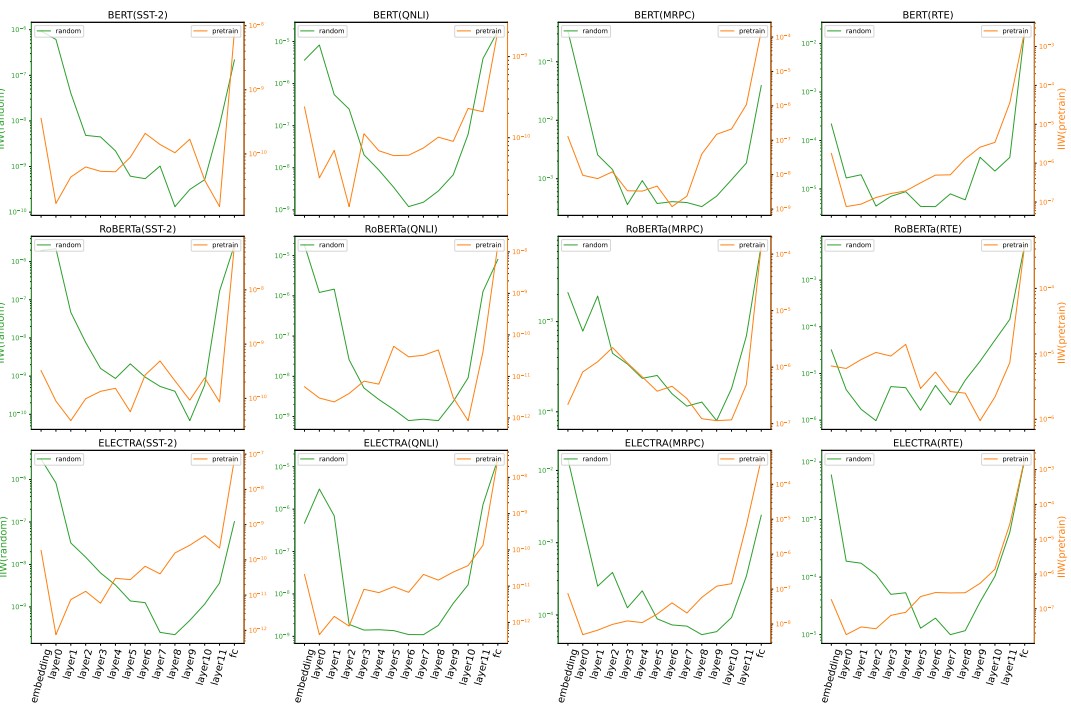

Figure 1: The IIW of pre-trained and non-pre-trained language models across different layers. The orange line represents the pre-trained language model, while the green line represents the non-pre-trained language model. The left y-axis in each graph shows the IIW scale for the pre-trained language model, while the right y-axis shows the scale for the non-pre-trained language model. Please note that the scale ranges on the left and right y-axes are not the same. The results of the remaining experiments are shown in Fig4.

## 3.3 IIW DURING FINE-TUNING PROCESS

Previous experiments have demonstrated that pre-trained language models tend to have lower IIW and better test performance. This naturally raises the question of whether IIW could serve as a proxy

for generalization performance, allowing us to monitor IIW during training as a direct reflection of the model's performance. One appealing aspect of using IIW as a proxy for generalization is that its evaluation does not rely on the test set. Specifically, we evaluated the IIW and test performance of pre-trained language models at the initial state and across each epoch during training. Ideally, we would expect that models with lower IIW would consistently perform better on the test set. However, as shown in Fig2, we observe that for both pre-trained and non-pre-trained language models, lower IIW does not always correspond to better test performance, though this is true for some datasets. Even more concerning is that, for RoBERTa on the RTE dataset, the pre-trained language model achieves its best performance when its IIW is at its highest.

The work of (Wang et al., 2022) suggests that there are two distinct phases during model training: a memorization phase and a forgetting phase. In the memorization phase, IIW monotonically increases, while in the forgetting phase, IIW monotonically decreases. Since epoch 0 does not belong to the actual training process, we exclude the values from this point. We first analyze the non-pre-trained language models in Fig1. The training process of these models does not exhibit the two-phase phenomenon. For example, in BERT-MRPC, IIW shows a monotonically increasing trend throughout training. In contrast, for pre-trained language models, we find that in most cases during fine-tuning, the training process follows the memorization and forgetting phases. However, in the case of RoBERTa-MRPC, we observe two distinct spikes in IIW. These experimental results suggest that the training process of pre-trained language models does not always adhere to the two-phase pattern of memorization and forgetting.

Another noteworthy point is that pre-trained language models generally have significantly lower initial IIW compared to randomly initialized models, although in the case of BERT-RTE, the initial IIW for the pre-trained language model is somewhat higher. This observation leads us to a bold hypothesis: ***if a model starts with lower IIW across most tasks, it is likely to exhibit better generalization performance after fine-tuning***.

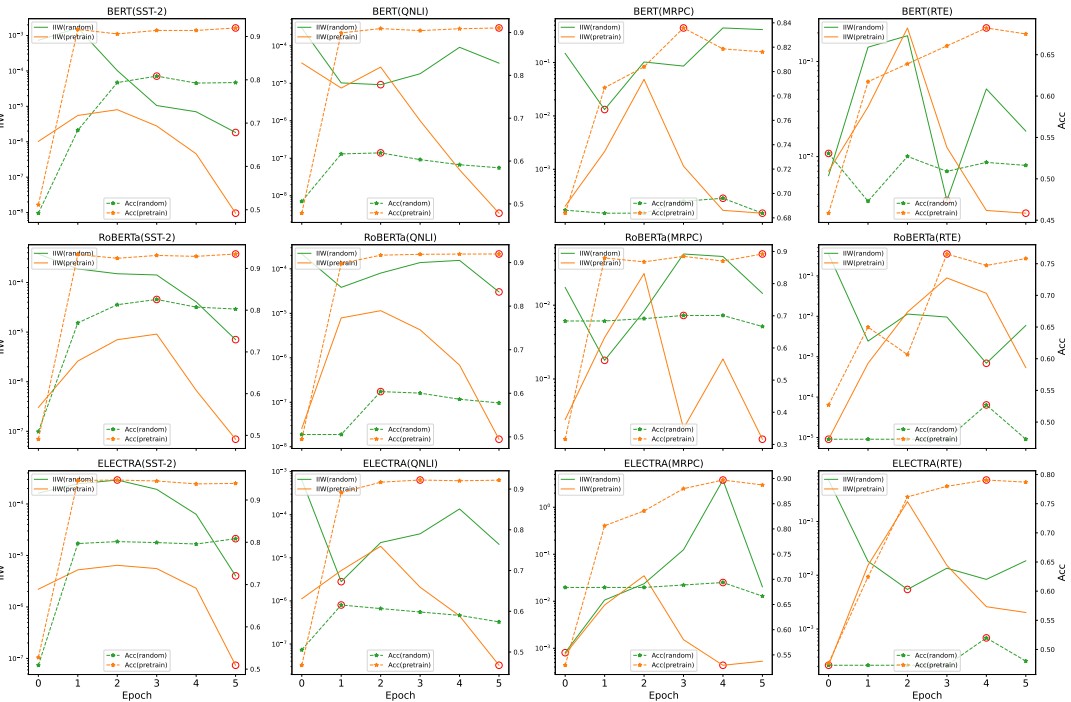

Figure 2: IIW and testset performance of pre-trained and non-pre-trained language models during the training process. Solid lines represent IIW values, while dashed lines represent test set performance. Orange indicates pre-trained language models, and green indicates non-pre-trained language models. For the dashed lines, we use circles to mark the highest test set performance. For the solid lines, we use circles to mark the lowest IIW. Note that Epoch=0 indicates the result before the models start training. The results of the remaining experiments are shown in Fig5.

## 3.4 IIW IN MODELS OF DIFFERENT FINE-TUNING METHODS

Previous experiments have shown that during fine-tuning, lower IIW does not consistently correlate with better generalization performance, whether for pre-trained or non-pre-trained language models. This suggests that IIW cannot serve as a reliable proxy for model generalization during training. However, earlier experiments also demonstrated that pre-trained language models generally exhibit lower IIW after training compared to non-pre-trained language models. In this experiment, we investigate whether IIW can serve as a generalization proxy for different models derived from pre-trained language models using various training methods. In other words, even with the same initialization, we explore whether models with lower IIW after fine-tuning tend to have better generalization performance.

(Kumar et al., 2022) proposed the linear probing then full fine-tuning (LP-FT) method, which initially freezes the feature extractor and trains only the head, followed by full fine-tuning of the entire model. We consider full fine-tuning and LP-FT as two distinct training methods and evaluate the performance and IIW of the resulting models. For LP-FT, we fine-tune only the randomly initialized head during the first epoch and then proceed with full-parameter training. The experimental results are shown in Table2. In some cases, models with lower IIW obtained through different fine-tuning methods indeed show better test performance. For example, this is observed in the two models fine-tuned on the RTE dataset with RoBERTa. However, in many other datasets, lower IIW does not consistently result in better test performance, as seen in BERT's results on RTE, RoBERTa's results on SST-2, and ELECTRA's results on MRPC.

Table 2: The IIW and test accuracy (Acc) of pre-trained language models across various datasets using different fine-tuning methods. "FT" represents full fine-tuning, while "LP-FT" stands for linear probing followed by full fine-tuning (Kumar et al., 2022). A green background indicates lower IIW, while a red background signifies higher IIW.

| Dataset | Metric | BERT | | RoBERTa | | ELECTRA | |
|---|---|---|---|---|---|---|---|
| | | FT | LP-FT | FT | LP-FT | FT | LP-FT |
| RTE | IIW | 2.54e-03 | 7.78e-03 | 5.37e-04 | 4.50e-04 | 2.03e-03 | 4.68e-03 |
| | Acc | 0.6751 | 0.6931 | 0.7581 | 0.8051 | 0.787 | 0.7942 |
| MRPC | IIW | 1.50e-04 | 3.55e-04 | 1.53e-04 | 2.03e-04 | 5.31e-04 | 1.04e-03 |
| | Acc | 0.8162 | 0.8456 | 0.8922 | 0.8848 | 0.8873 | 0.8971 |
| MNLI-m | IIW | 7.10e-09 | 6.13e-09 | 4.00e-09 | 5.60e-09 | 4.65e-09 | 3.17e-09 |
| | Acc | 0.8358 | 0.8392 | 0.862 | 0.8691 | 0.8744 | 0.8799 |
| MNLI-mm | IIW | 3.53e-09 | 6.16e-09 | 4.00e-09 | 5.60e-09 | 4.65e-09 | 3.17e-09 |
| | Acc | 0.8361 | 0.8422 | 0.8621 | 0.8678 | 0.8764 | 0.8817 |
| QNLI | IIW | 3.43e-09 | 1.58e-08 | 1.48e-08 | 7.47e-08 | 3.21e-08 | 3.32e-08 |
| | Acc | 0.9107 | 0.909 | 0.9196 | 0.9191 | 0.9218 | 0.9275 |
| QQP | IIW | 1.49e-09 | 1.95e-09 | 2.11e-09 | 6.39e-09 | 4.61e-09 | 5.23e-09 |
| | Acc | 0.9094 | 0.9103 | 0.9107 | 0.9123 | 0.9204 | 0.9204 |
| SST-2 | IIW | 9.60e-09 | 8.07e-08 | 6.92e-08 | 1.67e-07 | 7.37e-08 | 1.97e-07 |
| | Acc | 0.9197 | 0.9151 | 0.9346 | 0.9438 | 0.9392 | 0.9495 |

## 4 ANALYSIS AND DISCUSSION

**Experimental Observations.** Previous experiments have shown that IIW can serve as a proxy for generalization performance when comparing pre-trained language models with non-pre-trained language models. However, when IIW is applied during the fine-tuning process of pre-trained language models, its effectiveness as a generalization proxy diminishes. Furthermore, even when the same model is trained using different methods, IIW fails to consistently reflect generalization ability as a reliable proxy.

**Question.** Do the existing experimental results suggest that IIW itself cannot serve as a proxy for generalization? Or do they indicate that the current information-theoretic generalization bounds are ineffective at measuring the generalization ability of language models?

**Analysis.** Since we cannot directly optimize generalization error, existing methods construct an upper bound for it and then minimize this upper bound to reduce the generalization error as much as possible. However, one issue with minimizing the generalization error based on an upper bound is that if the given bound is not tight enough, it may not be directly comparable. For example, if we have $a \leq 0.3$ and $b \leq 0.2$, it does not necessarily mean that $b < a$ since $a$ could be 0.1 and $b$ could be 0.15. However, previous experiments have shown that IIW can distinguish between the performance of pre-trained and non-pre-trained language models, indicating that the current information-theoretic bounds do have some discriminatory power.

As for the inability of IIW to serve as a generalization proxy during the fine-tuning process, one possible reason could be that the current generalization bounds are not tight enough. Another, more likely reason is that the current method of calculating IIW may lack precision, leading to some degree of error. In fact, if the existing calculation method can only approximate IIW with some error, this could well explain the experimental observations. This would mean that when the performance difference between two models is significant i.e., there is a large difference in IIW—and the error in calculating IIW is smaller than this difference, IIW can serve as a proxy for generalization. However, when the performance difference between two models is small and the error in calculating IIW exceeds this difference, IIW can no longer serve as a generalization proxy.

**Theoretical Proof.** We have revisited the existing method for calculating IIW and discovered that there is indeed some degree of error in the precision estimation of the current IIW. Specifically, we present the following theoretical proof to illustrate this.

**Proposition 1.** *When $|\theta_Z - \theta_0| \leq \epsilon$, where $\epsilon > 0$ is an arbitrary small value, the current method for calculating IIW provides a good approximation. However, as $|\theta_Z - \theta_0|$ becomes larger, the precision of the IIW approximation decreases.*

**Empirical Evidence.** We calculated the probability distribution of $|\Delta\theta|$ for all parameters of the pre-trained language models after fine-tuning, along with the 1/2 and 3/4 quantiles of $|\Delta\theta|$. As shown in Fig3, across all datasets, the 1/2 quantile of $|\Delta\theta|$ exceeds $1 \times 10^{-4}$, and in some cases, even surpasses $1 \times 10^{-3}$. From Proposition1, we know that the IIW approximation is only accurate when $|\Delta\theta|$ is sufficiently small, and $1 \times 10^{-4}$ is not a negligible value. In cases where the 1/2 quantile of $|\Delta\theta|$ exceeds $1 \times 10^{-3}$, the approximation error of IIW is further amplified. Therefore, the experimental results indicate that the current IIW approximation method has significant inherent errors.

From the above analysis, it is clear that the current IIW evaluation method has inherent errors. To use IIW as a reliable proxy for generalization error, more precise calculation methods are needed. Only with sufficiently accurate IIW approximations can we properly assess whether the current generalization bounds are tight enough to serve as effective proxies during the training process. This also underscores the necessity of developing tighter and computationally feasible upper bounds for generalization.

## 5 RELATED WORK

**Pre-training model.** The traditional training paradigm in Natural Language Processing (NLP) involves using labeled data for specific tasks and designing models to fit this task-specific data. In contrast, the pre-training paradigm has emerged as a more effective approach. This method involves initially performing self-supervised learning on large amounts of raw text data, followed by fine-tuning on downstream tasks. GPT-1 (Radford et al., 2018) uses the Transformer's encoder to pre-train on large amounts of raw text data with a next token prediction task. GPT-2 (Radford et al.) significantly expanded the model size and training data, resulting in even more impressive text generation capabilities. GPT-3 (Brown et al., 2020) further increased the model's parameters to 175 billion, making it one of the largest and most powerful language models at the time. InstructGPT (Ouyang et al., 2022) uses Reinforcement Learning from Human Feedback (RLHF) to align the language model. Due to the closed-source nature of the GPT series, the open-source community has subsequently released many language models (Touvron et al., 2023a;b; Bai et al., 2023). BERT (Devlin et al., 2018) uses the Transformer's encoder to pre-train on masked token prediction and next sentence prediction tasks. RoBERTa (Liu et al., 2019) is an enhanced version of BERT, which optimizes the pre-training process using larger datasets, dynamic masking, and the removal of the Next Sentence Prediction task. XLNet (Yang et al., 2019) masks attention weights so that the input

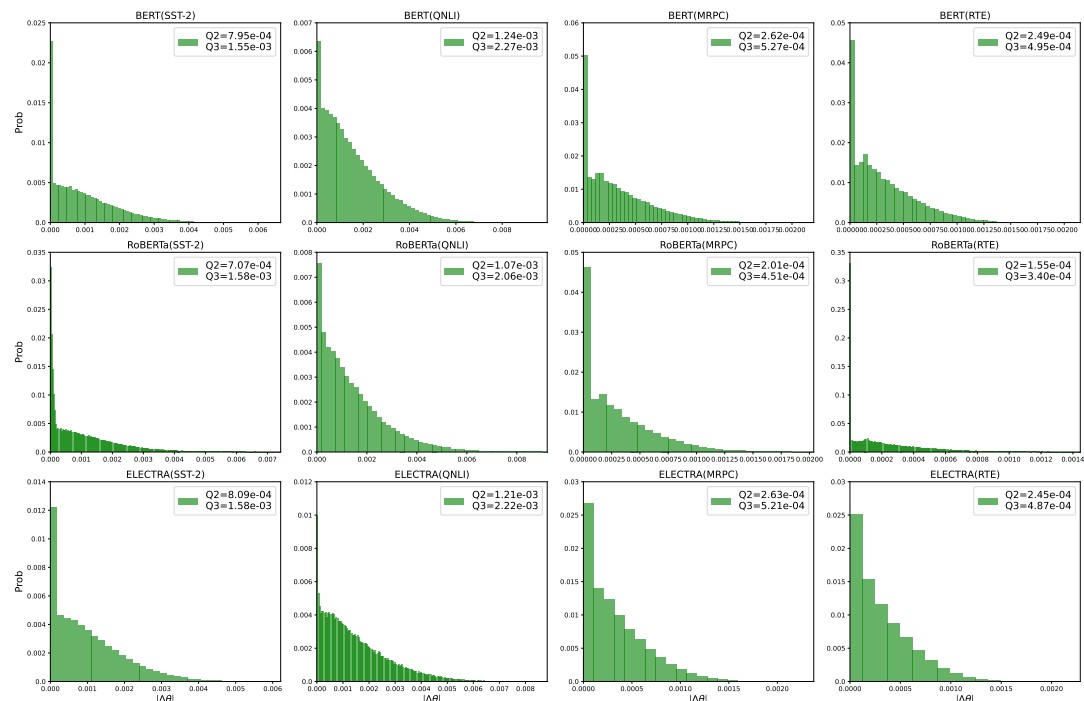

Figure 3: Distribution of $|\Delta\theta| \triangleq |\theta_Z - \theta_0|$ of the pre-trained language model on various datasets. $Q2$ represents 1/2 quantile of $|\Delta\theta|$, and $Q3$ represents 3/4 quantile of $|\Delta\theta|$. The results of the remaining experiments are shown in Fig6.

sequence is autoregressively generated in a random order. ELECTRA (Clark et al., 2020) uses a novel pre-training method involving replaced token detection rather than masked language modeling.

**Information Theory.** The information-theoretic generalization bound uses information theory to derive an upper bound on the generalization error. This bound not only enhances our understanding of generalization error but also provides insights for designing new learning algorithms. (Russo & Zou, 2016) show that the mutual information between the collection of empirical risks of the available hypotheses and the final output of the algorithm can be used effectively to analyze and control the bias in data. (Xu & Raginsky, 2017) derive upper bounds on the generalization error of a learning algorithm in terms of the mutual information between its input and output, which itself extended the results of (Russo & Zou, 2016) to a more general setting. A series of follow-ups tightened this bound and verified it is an effective measure of generalization capability of learning algorithms (Negrea et al., 2019; Wang & Mao, 2023). (Wang et al., 2022) propose an algorithm for the efficient approximation of IIW and we build an IIW-based information bottleneck.

## 6 CONCLUSION

In this work, we advance the application of information-theoretic generalization bounds in language models. Our findings reveal that pre-trained language models exhibit lower IIW compared to non-pre-trained language models, which may account for the superior effectiveness of pre-training. However, IIW proves inadequate as a generalization proxy during the fine-tuning process of these models. Furthermore, when different fine-tuning methods are applied to the same pre-trained language model, a lower IIW does not necessarily correlate with improved performance. This underscores the limitations of current IIW estimation method. We provide both theoretical and empirical evidence to highlight the shortcomings of existing approaches in accurately estimating IIW, emphasizing the need for more precise estimation techniques. A sufficiently accurate approximation of IIW is essential for determining whether the current generalization bounds are tight enough to reliably serve as a proxy for generalization during training.

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

APPENDIX

## A. PROOF OF PROPOSITION 1

**Proposition 1.** *When $|\theta_Z - \theta_0| \leq \epsilon$, where $\epsilon > 0$ is an arbitrary small value, the current method for calculating IIW provides a good approximation. However, as $|\theta_Z - \theta_0|$ becomes larger, the precision of the IIW approximation decreases.*

*Proof.* (Wang et al., 2022) assume both $p(W) = \mathcal{N}(W \mid \theta_0, \Sigma_0)$ and $p(W|Z) = \mathcal{N}(W \mid \theta_Z, \Sigma_Z)$ are Gaussian distributions. To simplify the computation, (Wang et al., 2022) further assume that the covariances of the prior and posterior are proportional. By using influence functions (Ling, 1984) and Poisson bootstrapping, $I(W; Z)$ can be expressed as:

$$I(W; Z) \propto n\mathbb{E}_{P_Z}[(\theta_Z - \theta_0)^T \mathbf{F}_\theta(\theta_Z - \theta_0)] \tag{8}$$

We follow the same assumptions as (Wang et al., 2022), where $p(W) \triangleq p(W; \theta_W) = \mathcal{N}(W \mid \theta_0, \Sigma_0)$ and $p(W|Z) \triangleq p(W|Z; \theta_{W|Z}) = \mathcal{N}(W \mid \theta_Z, \Sigma_Z)$ are both Gaussian distributions, with $\theta_W \triangleq (\theta_0, \Sigma_0)$ and $\theta_{W|Z} \triangleq (\theta_Z, \Sigma_Z)$.

We define $\Delta\theta \triangleq (\theta_0 - \theta_Z, \Sigma_0 - \Sigma_Z)$, then

$$\theta_W = \theta_{W|Z} + \Delta\theta \tag{9}$$

Since $p(W)$ and $p(W|Z)$ are both Gaussian distributions, $p(W|Z)$ can be expressed as

$$p(W; \theta_W) = p(W|Z; \theta_{W|Z} + \Delta\theta) \tag{10}$$

We then compute the KL divergence between $p(W)$ and $p(W|Z)$:

$$\mathrm{KL}(p(W|Z; \theta_{W|Z})\|p(W; \theta_W)) = \mathrm{KL}(p(W|Z; \theta_{W|Z})\|p(W|Z; \theta_{W|Z} + \Delta\theta)) \tag{11}$$

We further perform a second-order Taylor approximation on the above expression, yielding:

$$\mathrm{KL}(p(W|Z; \theta_{W|Z})\|p(W|Z; \theta_{W|Z} + \Delta\theta)) \approx \int p(W|Z; \theta_{W|Z}) \log p(W|Z; \theta_{W|Z}) dW \tag{12}$$

$$- \int p(W|Z; \theta_{W|Z})[\log p(W|Z; \theta_{W|Z}) + \left(\nabla \log p(W|Z; \theta_{W|Z})\right)^T \Delta\theta \tag{13}$$

$$+ \frac{1}{2}\Delta\theta^T \nabla^2 \log p(W|Z; \theta_{W|Z})\Delta\theta] dW \tag{14}$$

$$= \underbrace{\int p(W|Z; \theta_{W|Z}) \log \frac{p(W|Z; \theta_{W|Z})}{p(W|Z; \theta_{W|Z})} dW}_{\mathrm{KL}(p(W|Z;\theta_{W|Z})\|p(W|Z;\theta_{W|Z}))=0} - \underbrace{\left(\int \nabla p(W|Z; \theta_{W|Z}) dW\right)^T \Delta\theta}_{=0} \tag{15}$$

$$- \frac{1}{2}\Delta\theta^T \left(\int p(W|Z; \theta_{W|Z})\nabla^2 \log p(W|Z; \theta_{W|Z}) dW\right) \Delta\theta \tag{16}$$

For the second term, we have:

$$\int \nabla p(W|Z; \theta_{W|Z}) dW = \nabla \int p(W|Z; \theta_{W|Z}) dW = \nabla\mathbf{1} = 0 \tag{17}$$

For the third term, we have:

$$\nabla \log p(W|Z; \theta_{W|Z}) = \frac{\nabla p(W|Z; \theta_{W|Z})}{p(W|Z; \theta_{W|Z})} \tag{18}$$

$$\nabla^2 \log p(W|Z; \theta_{W|Z}) = \frac{\nabla^2 p(W|Z; \theta_{W|Z})}{p(W|Z; \theta_{W|Z})} - \frac{\nabla p(W|Z; \theta_{W|Z}) \nabla p(W|Z; \theta_{W|Z})^T}{p(W|Z; \theta_{W|Z}) p(W|Z; \theta_{W|Z})} \tag{19}$$

$$= \frac{\nabla^2 p(W|Z; \theta_{W|Z})}{p(W|Z; \theta_{W|Z})} - \nabla \log p(W|Z; \theta_{W|Z}) \nabla \log p(W|Z; \theta_{W|Z})^T \tag{20}$$

We can further obtain:

$$\mathrm{KL}(p(W|Z; \theta_{W|Z}) || p(W|Z; \theta_{W|Z} + \Delta\theta)) \tag{21}$$

$$\approx -\frac{1}{2} \Delta\theta^T \left( \int p(W|Z; \theta_{W|Z}) \nabla^2 \log p(W|Z; \theta_{W|Z}) dW \right) \Delta\theta \tag{22}$$

$$= -\frac{1}{2} \Delta\theta^T \underbrace{\left( \int \nabla^2 p(W|Z; \theta_{W|Z}) dW \right)}_{=0} \Delta\theta \tag{23}$$

$$+ \frac{1}{2} \Delta\theta^T \underbrace{\left( \int p(W|Z; \theta_{W|Z}) \log p(W|Z; \theta_{W|Z}) \nabla \log p(W|Z; \theta_{W|Z})^T dW \right)}_{F_{\theta_{W|Z}}} \Delta\theta \tag{24}$$

$$= \frac{1}{2} \Delta\theta^T F_{\theta_{W|Z}} \Delta\theta \tag{25}$$

For the first term in the above equation, we have:

$$\int \nabla^2 p(W|Z; \theta_{W|Z}) dW = \nabla^2 \int p(W|Z; \theta_{W|Z}) dW = \nabla^2 \mathbf{1} = 0 \tag{26}$$

Based on the above derivation, we can also obtain an approximate result for IIW, which is equivalent to the calculation in Eq19.

$$I(W; Z) = \mathbb{E}_{P_Z}[\mathrm{KL}(p(W|Z) \parallel p(W))] \propto \frac{1}{2} \mathbb{E}_{P_Z}[\Delta\theta^T F_{\theta_{W|Z}} \Delta\theta] \tag{27}$$

Therefore, we can understand that the current method for calculating IIW (Wang et al., 2022) is based on the second-order Taylor approximation of $\mathrm{KL}(p(W|Z) \parallel p(W))$. However, Taylor approximation only provides good accuracy when $\Delta\theta$ is sufficiently small, i.e., $\Delta\theta \leq \epsilon$. When $\Delta\theta$ is larger, the approximation error increases.

$\square$

# B. IIW OF PRE-TRAINED LANGUAGE MODEL AND NON-PRE-TRAINED LANGUAGE MODEL

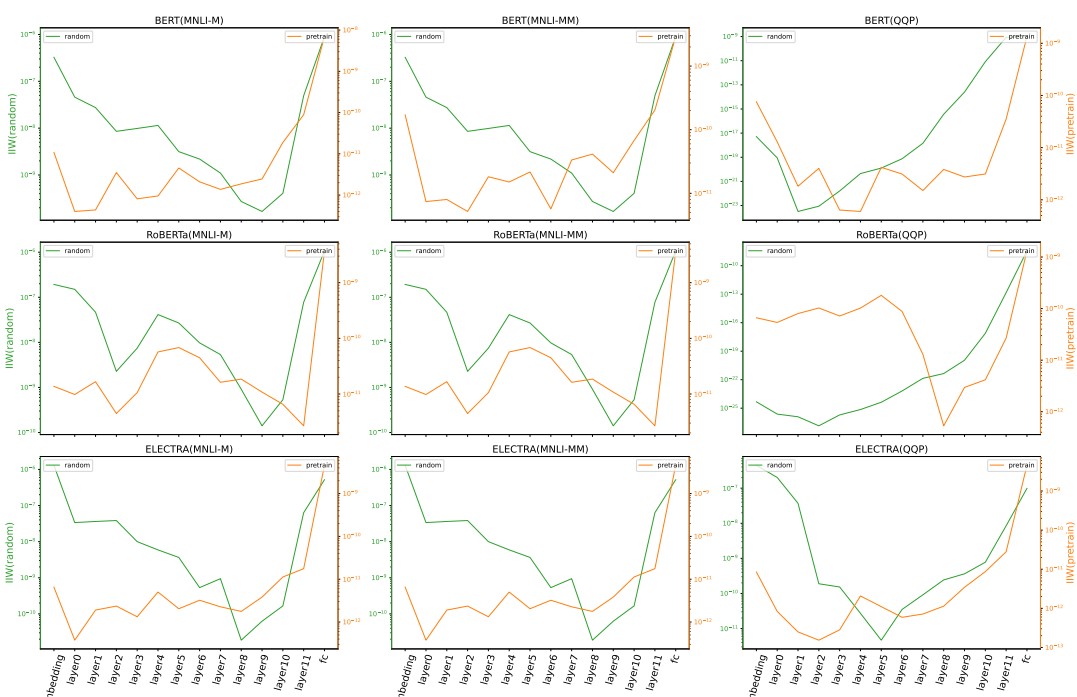

Figure 4: The IIW of pre-trained and non-pre-trained language models across different layers. The orange line represents the pre-trained language model, while the green line represents the non-pre-trained language model. The left y-axis in each graph shows the IIW scale for the pre-trained language model, while the right y-axis shows the scale for the non-pre-trained language model. Please note that the scale ranges on the left and right y-axes are not the same.

## C. IIW DURING FINE-TUNING PROCESS

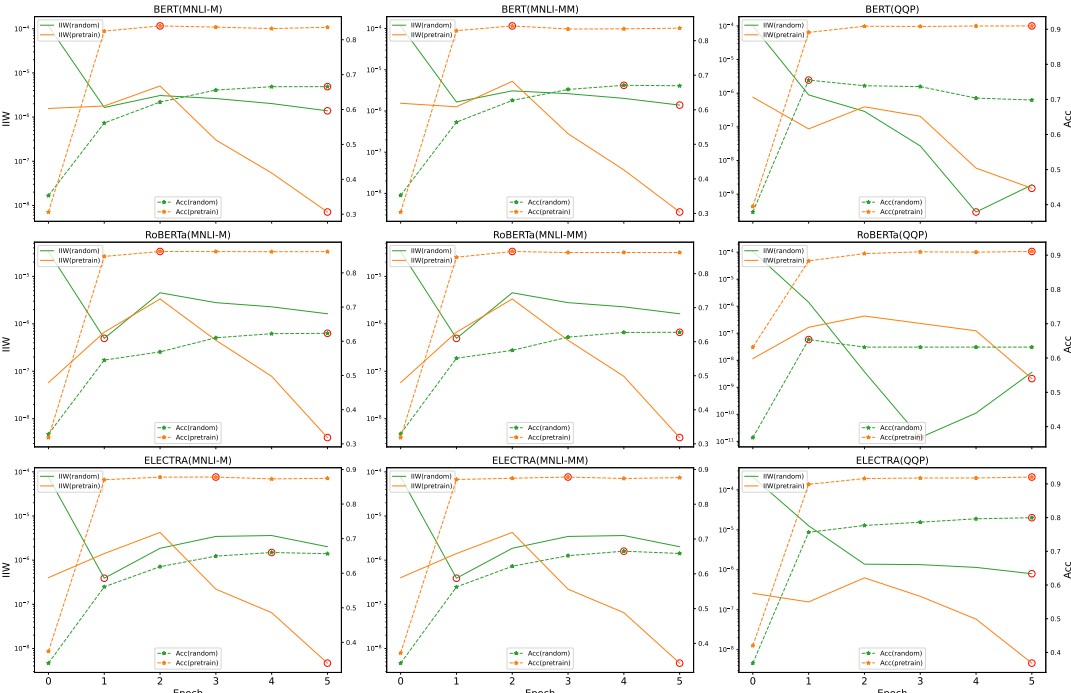

Figure 5: IIW and testset performance of pre-trained and non-pre-trained language models during the training process. Solid lines represent IIW values, while dashed lines represent test set performance. Orange indicates pre-trained language models, and green indicates non-pre-trained language models. For the dashed lines, we use circles to mark the highest test set performance. For the solid lines, we use circles to mark the lowest IIW. Note that Epoch=0 indicates the result before the models start training.

## D. WEIGHT GAP DISTRIBUTION

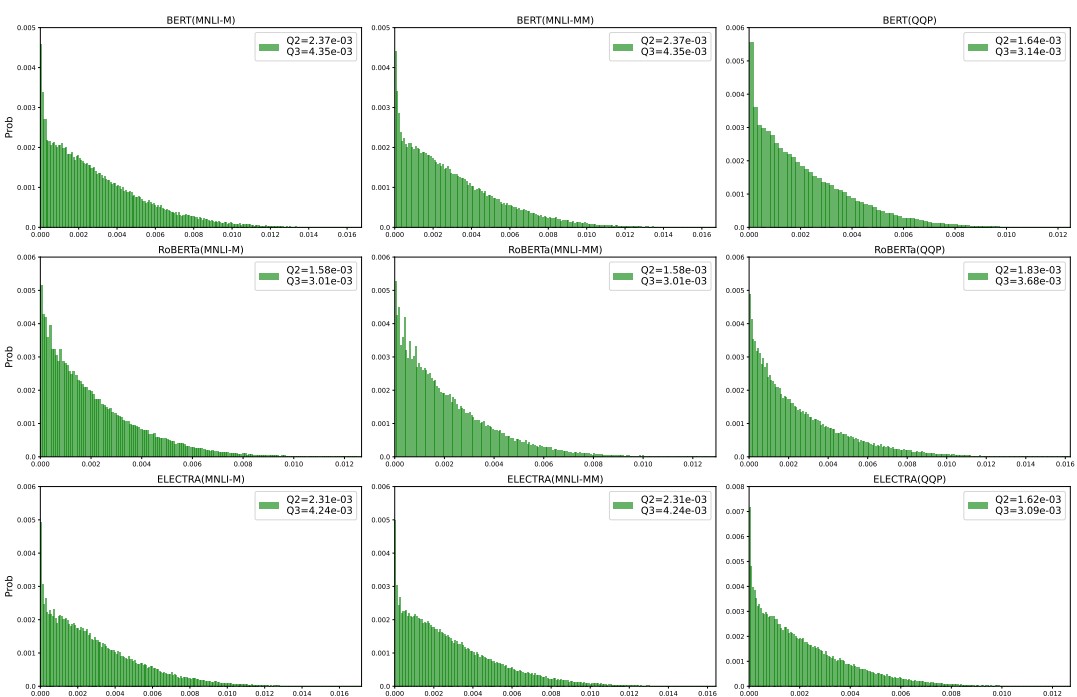

Figure 6: Distribution of $|\Delta\theta| \triangleq |\theta_Z - \theta_0|$ of the pre-trained language model on various datasets. $Q2$ represents 1/2 quantile of $|\Delta\theta|$, and $Q3$ represents 3/4 quantile of $|\Delta\theta|$.

# E. IIW of each layer in pre-trained language model

Table 3: Layer IIW of BERT on various datasets.

| | BERT | | | | | | |
| | SST-2 | QNLI | MRPC | RTE | MNLI-M | MNLI-MM | QQP |
|---|---|---|---|---|---|---|---|
| embedding | 3.59e-10 | 2.39e-10 | 1.28e-07 | 1.78e-06 | 1.08e-11 | 1.71e-10 | 7.56e-11 |
| layer0 | 1.69e-11 | 3.19e-11 | 9.61e-09 | 7.62e-08 | 4.01e-13 | 7.42e-12 | 1.29e-11 |
| layer1 | 4.37e-11 | 6.96e-11 | 7.67e-09 | 8.83e-08 | 4.37e-13 | 8.00e-12 | 1.82e-12 |
| layer2 | 6.27e-11 | 1.42e-11 | 1.21e-08 | 1.30e-07 | 3.51e-12 | 5.18e-12 | 4.00e-12 |
| layer3 | 5.36e-11 | 1.12e-10 | 3.41e-09 | 1.65e-07 | 8.12e-13 | 1.82e-11 | 6.37e-13 |
| layer4 | 5.28e-11 | 6.91e-11 | 3.33e-09 | 1.92e-07 | 9.51e-13 | 1.51e-11 | 5.97e-13 |
| layer5 | 8.88e-11 | 6.02e-11 | 4.66e-09 | 3.12e-07 | 4.54e-12 | 2.17e-11 | 4.18e-12 |
| layer6 | 2.10e-10 | 6.09e-11 | 1.18e-09 | 4.92e-07 | 2.09e-12 | 5.70e-12 | 3.12e-12 |
| layer7 | 1.40e-10 | 7.48e-11 | 2.38e-09 | 5.00e-07 | 1.38e-12 | 3.34e-11 | 1.51e-12 |
| layer8 | 1.05e-10 | 1.02e-10 | 3.97e-08 | 1.28e-06 | 1.87e-12 | 4.14e-11 | 3.82e-12 |
| layer9 | 1.69e-10 | 9.02e-11 | 1.49e-07 | 2.58e-06 | 2.46e-12 | 2.11e-11 | 2.74e-12 |
| layer10 | 3.90e-11 | 2.30e-10 | 2.12e-07 | 3.42e-06 | 1.88e-11 | 6.77e-11 | 3.12e-12 |
| layer11 | 1.51e-11 | 2.09e-10 | 1.08e-06 | 3.58e-05 | 8.87e-11 | 2.04e-10 | 3.52e-11 |
| fc | 8.24e-09 | 2.06e-09 | 1.49e-04 | 2.49e-03 | 6.96e-09 | 2.91e-09 | 1.34e-09 |

Table 4: Layer IIW of RoBERTa on various datasets.

| | RoBERTa | | | | | | |
| | SST-2 | QNLI | MRPC | RTE | MNLI-M | MNLI-MM | QQP |
|---|---|---|---|---|---|---|---|
| embedding | 3.26e-10 | 5.64e-12 | 2.20e-07 | 6.52e-06 | 1.37e-11 | 1.37e-11 | 6.63e-11 |
| layer0 | 8.89e-11 | 3.02e-12 | 8.24e-07 | 5.97e-06 | 9.80e-12 | 9.80e-12 | 5.39e-11 |
| layer1 | 3.88e-11 | 2.43e-12 | 1.25e-06 | 8.17e-06 | 1.67e-11 | 1.67e-11 | 8.01e-11 |
| layer2 | 9.85e-11 | 3.88e-12 | 2.22e-06 | 1.05e-05 | 4.47e-12 | 4.47e-12 | 1.03e-10 |
| layer3 | 1.35e-10 | 7.68e-12 | 1.18e-06 | 9.25e-06 | 1.06e-11 | 1.06e-11 | 7.18e-11 |
| layer4 | 1.53e-10 | 6.52e-12 | 6.74e-07 | 1.40e-05 | 5.70e-11 | 5.70e-11 | 1.02e-10 |
| layer5 | 5.67e-11 | 5.30e-11 | 3.76e-07 | 2.95e-06 | 6.83e-11 | 6.83e-11 | 1.81e-10 |
| layer6 | 2.70e-10 | 2.96e-11 | 4.59e-07 | 5.27e-06 | 4.48e-11 | 4.48e-11 | 8.81e-11 |
| layer7 | 4.89e-10 | 3.23e-11 | 2.79e-07 | 2.65e-06 | 1.63e-11 | 1.63e-11 | 1.30e-11 |
| layer8 | 2.11e-10 | 4.34e-11 | 1.22e-07 | 2.51e-06 | 1.85e-11 | 1.85e-11 | 5.33e-13 |
| layer9 | 9.27e-11 | 3.09e-12 | 1.13e-07 | 9.53e-07 | 1.08e-11 | 1.08e-11 | 2.96e-12 |
| layer10 | 2.39e-10 | 8.62e-13 | 1.17e-07 | 2.19e-06 | 6.54e-12 | 6.54e-12 | 4.16e-12 |
| layer11 | 8.57e-11 | 3.91e-11 | 4.93e-07 | 7.36e-06 | 2.68e-12 | 2.68e-12 | 2.68e-11 |
| fc | 6.69e-08 | 1.45e-08 | 1.44e-04 | 4.59e-04 | 3.72e-09 | 3.72e-09 | 1.31e-09 |

Table 5: Layer IIW of ELECTRA on various datasets.

| | ELECTRA | | | | | | |
| | SST-2 | QNLI | MRPC | RTE | MNLI-M | MNLI-MM | QQP |
|---|---|---|---|---|---|---|---|
| embedding | 1.85e-10 | 2.10e-11 | 7.41e-08 | 1.83e-07 | 6.52e-12 | 6.52e-12 | 8.49e-12 |
| layer0 | 7.54e-13 | 4.60e-13 | 4.97e-09 | 1.80e-08 | 3.77e-13 | 3.77e-13 | 8.34e-13 |
| layer1 | 7.46e-12 | 1.47e-12 | 6.68e-09 | 3.00e-08 | 1.90e-12 | 1.90e-12 | 2.50e-13 |
| layer2 | 1.28e-11 | 7.91e-13 | 9.79e-09 | 2.67e-08 | 2.36e-12 | 2.36e-12 | 1.53e-13 |
| layer3 | 5.87e-12 | 8.22e-12 | 1.23e-08 | 6.44e-08 | 1.33e-12 | 1.33e-12 | 2.82e-13 |
| layer4 | 3.00e-11 | 6.63e-12 | 1.10e-08 | 7.90e-08 | 5.02e-12 | 5.02e-12 | 2.07e-12 |
| layer5 | 2.74e-11 | 9.67e-12 | 1.93e-08 | 2.24e-07 | 2.04e-12 | 2.04e-12 | 1.11e-12 |
| layer6 | 6.58e-11 | 6.76e-12 | 4.04e-08 | 2.92e-07 | 3.23e-12 | 3.23e-12 | 5.89e-13 |
| layer7 | 4.00e-11 | 2.11e-11 | 2.07e-08 | 2.85e-07 | 2.27e-12 | 2.27e-12 | 7.15e-13 |
| layer8 | 1.57e-10 | 1.47e-11 | 5.77e-08 | 2.88e-07 | 1.77e-12 | 1.77e-12 | 1.14e-12 |
| layer9 | 2.60e-10 | 2.45e-11 | 1.21e-07 | 5.38e-07 | 3.82e-12 | 3.82e-12 | 3.45e-12 |
| layer10 | 4.87e-10 | 3.69e-11 | 1.43e-07 | 1.37e-06 | 1.13e-11 | 1.13e-11 | 8.67e-12 |
| layer11 | 2.16e-10 | 1.34e-10 | 6.74e-06 | 2.72e-05 | 1.76e-11 | 1.76e-11 | 2.79e-11 |
| fc | 7.22e-08 | 3.19e-08 | 5.24e-04 | 2.00e-03 | 4.59e-09 | 4.59e-09 | 4.55e-09 |

