# OpenReview forum: "Can Information-Theoretic Generalization Bound Explain the Generalization of Pre-trained Language Model?"
_ICLR.cc/2025/Conference — ICLR 2025 Conference Withdrawn Submission_

### Official Review · Reviewer_aaJ2 · 2024-10-27

**Soundness:** 1
**Presentation:** 3
**Contribution:** 1
**Rating:** 3
**Confidence:** 3

**Summary:**

**Main contributions of this paper:**

- Information stored in the weights (IIW) can serve as a proxy for generalization error in pre-trained models but not during fine-tuning.
- The main reason is “some degree of error in precision estimation IIW”

Models: BERT-base, RoBERTa-base,ELECTRA-base

**Strengths:**

This paper empirically demonstrates the findings on encoder-style language models: Information stored in the weights (IIW) can serve as a proxy for generalization error in pre-trained models but not during fine-tuning.

**Weaknesses:**

1. Limited verification: this paper focuses on pre-trained language models, but they only used encoder-style models. Most commonly used pre-trained models today are decoder-only, and this has not been validated in this paper. This issue also extends to tasks such as language modeling and other sequence-to-sequence tasks.
2. The contribution is marginal. Methods in Sec. 2 are all from previous papers. What is new from you? The theoretical analysis in Sec. 4 is lack of experiments verification.
3. Although the experimental analysis identified precision error in IIW for improvement, it did not provide any solutions. I believe these insights  can be figured out through extensive experiments, but the most crucial aspect is how to solve this issue, which the paper does not address.

**Questions:**

See in Weaknesses.

---

> ### Author Response · Authors · 2024-11-24
> **Rebuttal by Authors**
>
> Thank you very much for your careful review. We appreciate your feedback and will do our best to answer your questions.
>
> ---
>
> Q: Limited verification: this paper focuses on pre-trained language models, but they only used encoder-style models. Most commonly used pre-trained models today are decoder-only, and this has not been validated in this paper. This issue also extends to tasks such as language modeling and other sequence-to-sequence tasks.
>
> A: We have added the experimental results for GPT-2 (Medium). The training performance, test performance, generalization error, and IIW of the pre-trained GPT-2 model and non-pre-trained model are shown in the table below. It is evident that pre-trained models consistently exhibit lower IIW while achieving better test performance.
>
> |    GPT2       | train_acc | test_acc | acc_gap | iiw      |
> | ------------- | --------- | -------- | ------- | -------- |
> |RTE(pretrain)|0.9964|0.6606|0.3358|2.08e-03|
> |RTE(random)|0.9652|0.4693|0.4959|1.55e-02|
> |QNLI(pretrain)|0.9998|0.9109|0.08898|3.96e-09|
> |QNLI(random)|0.9617|0.5795|0.3822|2.17e-05|
> |QQP(pretrain)|0.9997|0.9075|0.09212|3.44e-10|
> |QQP(random)|0.9787|0.7986|0.1801|1.04e-06|
> |SST2(pretrain)|0.9985|0.9358|0.0627|3.36e-08|
> |SST2(random)|0.983|0.7982|0.1848|3.94e-06|
> |MNLI(pretrain)|0.9994|0.8446|0.1548|1.01e-09|
> |MNLI(random)|0.8156|0.588|0.2276|9.02e-06|
> |MNLI_MIS(pretrain)|0.9994|0.8453|0.1541|9.43e-10|
> |MNLI_MIS(random)|0.8096|0.598|0.2116|5.28e-06|
> |MRPC(pretrain)|0.9988|0.8211|0.1777|1.18e-04|
> |MRPC(random)|0.9936|0.6593|0.3343|3.76e-03|
>
>
> We observe that during fine-tuning, a lower IIW does not always correspond to the best test performance, which aligns with the conclusions of our paper. The following table shows the training performance, test performance, and generalization error of GPT-2 during the fine-tuning process on the RTE dataset. Notably, GPT-2 achieves its best test performance at epoch 2, but this does not correspond to the lowest IIW.
>
> |    RTE(GPT2)       | train_acc | test_acc | acc_gap | iiw      |
> | ------------- | --------- | -------- | ------- | -------- |
> |Epoch 0|0.4975|0.4729|0.02462|2.35e+01|
> |Epoch 1|0.5533|0.556|0.0026|7.06e-02|
> |Epoch 2|0.8282|0.6643|0.1639|6.63e-01|
> |Epoch 3|0.9366|0.6426|0.294|3.94e-01|
> |Epoch 4|0.9732|0.6606|0.3126|2.49e-02|
> |Epoch 5|0.9964|0.6606|0.3358|2.08e-03|
>
> We also have added the experimental results of GPT-2 on the XSum and CNN/DailyMail summarization tasks. Perplexity (PPL) is used as the evaluation metric. We observe that pre-trained models achieve lower PPL while also exhibiting lower IIW.
>
> |    GPT2       | train_ppl | test_ppl | ppl_gap | iiw      |
> | ------------- | --------- | -------- | ------- | -------- |
> |xsum(pretrain)|2.321|8.055|5.734|2.10e-01|
> |xsum(random)|59.98|127.9|67.95|1.08e+01|
> |cnn_dailymail(pretrain)|2.023|6.623|4.6|1.61e-01|
> |cnn_dailymail(random)|69.19|278.6|209.4|9.40e+00|
>
> The following are the results of GPT-2 during the fine-tuning process on the CNN/DailyMail dataset. It can be observed that the model achieves the smallest generalization error at Epoch 1, but the corresponding IIW is not the smallest. This is consistent with the conclusions of our previous experiments.
>
> |    cnn_dailymail       | train_ppl | test_ppl | ppl_gap | iiw      |
> | ------------- | --------- | -------- | ------- | -------- |
> |Epoch 0|13.53|11.21|2.32|1.31e+00|
> |Epoch 1|4.002|5.503|1.501|1.74e-02|
> |Epoch 2|3.264|5.564|2.3|5.07e-02|
> |Epoch 3|2.729|5.834|3.105|1.05e-01|
> |Epoch 4|2.313|6.271|3.958|1.41e-01|
> |Epoch 5|2.023|6.623|4.6|1.61e-01|

---

> > ### Author Response · Authors · 2024-11-24
> > **Rebuttal by Authors**
> >
> > Q: The contribution is marginal. Methods in Sec. 2 are all from previous papers. What is new from you? The theoretical analysis in Sec. 4 is lack of experiments verification.
> >
> > A: Our paper found that compared to randomly initialized models, pre-trained language models have lower IIW, which can explain the effectiveness of pre-trained language models from the perspective of information-theoretic generalization bounds. We also discovered the limitations of existing IIW approximation methods, namely that when the generalization error is relatively close, IIW cannot stably act as a proxy for the generalization error. These experimental phenomena are all new.
> >
> > In theory, we provide a new IIW proof method that is consistent with existing results, and our method shows that as the weight gap increases, the calculation error of IIW also increases. From the experiment, we calculated the 1/2 and 3/4 quantiles of the weight gap to validate our theory.
> >
> > ---
> >
> > Q: Although the experimental analysis identified precision error in IIW for improvement, it did not provide any solutions. I believe these insights can be figured out through extensive experiments, but the most crucial aspect is how to solve this issue, which the paper does not address.
> >
> > A: We believe that the effectiveness of using information theory to explain pre-trained models is a promising direction. However, there has been no prior work that combines the information-theoretic generalization bounds with pre-trained language models. Additionally, there has been no work that provides a theoretical explanation of the limitations of the current IIW approximation. While we have not solved the limitation problem of the IIW approximation, our work promotes the application of information theory generalization boundaries in pre-trained models.

---

> > > ### Comment · Reviewer_aaJ2 · 2024-11-27
> > >
> > > Thank you for considerable results and detailed classification.
> > > Partial my concerns have been addressed. However, the most important point (to me) is to figure out how to solve "errors in IIW approximation", which is a really important contribution. Currently, you have found the trends and analyzed the problem, but this paper is not complete. The most important part is left for future work.
> > >
> > > **In summary, I think this paper is not accomplished (for now). So I will keep score as it is.** If you fix these problems, I think it might be a great work. I am looking forward to your continual works.
> > >
> > > I have some advice for paper improvement as follows:
> > > For presentation:
> > > - you should directly point out the your analysis trend using a figure in front of your content.
> > > - then using the plain and direct word convey your results which is easy for reader to quick what have you done.
> > > For experiments:
> > > - you should be more closely with current framework, which help you to promote your impact.
> > > - like pre-training models, data generalization bounds, and information theory, and so on.
> > >
> > > Thank you once again for your rebuttal.

---

> > > > ### Author Response · Authors · 2024-11-27
> > > > **Rebuttal by Authors**
> > > >
> > > > We sincerely appreciate your response. We believe that significantly reducing the approximation error of IIW is not straightforward. During the learning process, the model's weights need to change to fit the training set, and these weight gaps are the fundamental cause of IIW approximation errors. Although a lower IIW does not always correspond to the best test performance, we find that using IIW as a model selection criterion is effective. Specifically, compare using the lowest IIW as the model selection criterion against the best test performance. We find that the average difference between using IIW as the selection criterion and the best test performance is only 0.0053. This suggests that, although lower IIW does not always correspond to better test performance, using IIW as a model selection criterion is entirely acceptable in practical applications.
> > > >
> > > > |Stats Info|Test Acc Gap|
> > > > | ------------- | --------- |
> > > > |mean|0.0053|
> > > > |medium|0.0029|
> > > > |max|0.0220|
> > > >
> > > > We appreciate your suggestions for improving the paper, and we will make the revisions accordingly.

---

### Official Review · Reviewer_MWY2 · 2024-11-01

**Soundness:** 3
**Presentation:** 2
**Contribution:** 2
**Rating:** 3
**Confidence:** 3

**Summary:**

This paper explores whether information-theoretic generalization bounds, particularly information stored in the weights (IIW), can explain the generalization capabilities of pretrained language models compared to non-pretrained ones. The authors aim to assess if IIW can serve as a reliable measure for generalization performance, especially during the fine-tuning of these models.

The motivation of this work comes from the remarkable success of pretrained LMs (such as BERT, RoBERTa, and ELECTRA) in various NLP tasks. But, the reasons for their good generalization performance is still unclear. Recent studies in information theory suggest that IIW, which measures the amount of information stored in a model's weights, may be a factor influencing generalization. Models with lower IIW are thought to have better generalization performance due to a smaller upper bound on generalization error. However, it is unknown whether this theory holds true for pre-trained language models. Given that, in this study, the authors investigate two main questions: 1) Can IIW explain why pre-trained language models generalize better than non-pre-trained models? and 2) Can IIW serve as a reliable proxy for generalization performance during the fine-tuning process?

To answer these questions, the authors conduct experiments comparing the IIW and performance of pretrained and non-pretrained LMs across various datasets. They also explore how IIW behaves during fine-tuning and whether it correlates with test performance.

**Strengths:**

- The exploration of IIW in the context of language models like BERT, RoBERTa, and ELECTRA fills an important gap in understanding the generalization capabilities of these models. The authors contribute by providing empirical evidence that pretrained models have lower IIW compared to non-pretrained models, suggesting that IIW can explain why pretrained models generalize better. This is a significant step forward in applying theoretical concepts to language models.

- The paper goes beyond aggregate IIW values, and provides a layer-specific analysis of IIW for both pretrained and non-pretrained models. This level of granularity helps uncover important insights into how different layers contribute to overall model generalization. For instance, they observe that the fully connected (fc) layer in pretrained models has significantly higher IIW than other layers, which they attribute to its role in adapting to downstream tasks.

- The authors critically assess the limitations of current methods for estimating IIW, arguing that these methods may lack precision and could explain why IIW does not consistently correlate with generalization performance during fine-tuning. The paper opens up new direction for future research aimed at improving these methods.

**Weaknesses:**

- In my opinion the main weakness of this study is about one of the main findings that shows IIW does not reliably serve as a proxy for generalization performance during fine-tuning. In some cases, lower IIW corresponds to worse test performance. This undermines one of the key goals of the paper -- using IIW as a reliable measure for generalization during fine-tuning. The inconsistency between IIW and test performance suggests that either IIW is not an adequate measure for complex models like pretrained language models or that current methods for calculating it are flawed. This weakens the overall impact of their findings since they cannot offer a practical solution for using IIW as a generalization proxy during fine-tuning.

- The authors explore different fine-tuning methods (e.g., full fine-tuning vs linear probing followed by full fine-tuning) but find that lower IIW does not consistently correlate with better test performance across these methods. The lack of consistent results across fine-tuning methods makes it difficult to draw actionable conclusions for practitioners who might want to use IIW as part of their model evaluation process.

- Without offering practical solutions or next steps for improving IIW estimation, the paper leaves readers with more questions than answers regarding how to apply its findings in real-world scenarios. This limits its usefulness for both researchers and practitioners looking for concrete takeaways.

- The paper mostly focuses on presenting empirical results without clearly testing specific hypotheses about why certain phenomena occur (e.g., why lower IIW does not always correlate with better test performance). A more hypothesis-driven approach could have strengthened the paper by providing clearer explanations or predictions about when and why certain patterns would emerge.

**Questions:**

- It is not so clear if these findings are generalizable for different architectures, for example for state-of-the-art autoregressive models. If the authors can provide some insights about the generalizability of their approach, that would greatly help to the readers.

- I think more explanation is needed for the inconsistency between IIW and test performance.

---

> ### Author Response · Authors · 2024-11-24
> **Rebuttal by Authors**
>
> Thank you very much for your careful review. We appreciate your feedback and will do our best to answer your questions.
>
> ---
>
> Q: In my opinion the main weakness of this study is about one of the main findings that shows IIW does not reliably serve as a proxy for generalization performance during fine-tuning. ...
>
> The authors explore different fine-tuning methods (e.g., full fine-tuning vs linear probing followed by full fine-tuning) but find that lower IIW does not consistently correlate with better test performance across these methods. The lack of consistent results across fine-tuning methods makes it difficult to draw actionable conclusions for practitioners who might want to use IIW as part of their model evaluation process.
>
> Without offering practical solutions or next steps for improving IIW estimation, the paper leaves readers with more questions than answers regarding how to apply its findings in real-world scenarios. This limits its usefulness for both researchers and practitioners looking for concrete takeaways.
>
> A: Since these questions are quite similar, we will provide a unified answer. We believe that the effectiveness of using information theory to explain pre-trained models is a promising direction. However, there has been no prior work that combines the information-theoretic generalization bounds with pre-trained language models. Additionally, there has been no work that provides a theoretical explanation of the limitations of the current IIW approximation. While we have not solved the limitation problem of the IIW approximation, our work promotes the application of information theory generalization boundaries in pre-trained models.
>
> ---
>
> Q: The paper mostly focuses on presenting empirical results without clearly testing specific hypotheses about why certain phenomena occur (e.g., why lower IIW does not always correlate with better test performance). A more hypothesis-driven approach could have strengthened the paper by providing clearer explanations or predictions about when and why certain patterns would emerge.
> I think more explanation is needed for the inconsistency between IIW and test performance.
>
> A: In fact, our experiments have shown that for models with significant differences in generalization error, such as pre trained and non pre trained models, IIW can serve as a proxy for generalization error. However, when the performance between models is relatively close, IIW cannot stably act as a proxy for generalization error. In Section 4, we provide relevant theories and experiments for verification. Our theory suggests that existing IIW approximation methods experience an increase in approximation error as the weight gap increases.
>
> ---

---

> > ### Author Response · Authors · 2024-11-24
> > **Rebuttal by Authors**
> >
> > Q: It is not so clear if these findings are generalizable for different architectures, for example for state-of-the-art autoregressive models. If the authors can provide some insights about the generalizability of their approach, that would greatly help to the readers.
> >
> > A: We have added the experimental results for GPT-2 (Medium). The training performance, test performance, generalization error, and IIW of the pre-trained GPT-2 model and non-pre-trained model are shown in the table below. It is evident that pre-trained models consistently exhibit lower IIW while achieving better test performance.
> >
> > |    GPT2       | train_acc | test_acc | acc_gap | iiw      |
> > | ------------- | --------- | -------- | ------- | -------- |
> > |RTE(pretrain)|0.9964|0.6606|0.3358|2.08e-03|
> > |RTE(random)|0.9652|0.4693|0.4959|1.55e-02|
> > |QNLI(pretrain)|0.9998|0.9109|0.08898|3.96e-09|
> > |QNLI(random)|0.9617|0.5795|0.3822|2.17e-05|
> > |QQP(pretrain)|0.9997|0.9075|0.09212|3.44e-10|
> > |QQP(random)|0.9787|0.7986|0.1801|1.04e-06|
> > |SST2(pretrain)|0.9985|0.9358|0.0627|3.36e-08|
> > |SST2(random)|0.983|0.7982|0.1848|3.94e-06|
> > |MNLI(pretrain)|0.9994|0.8446|0.1548|1.01e-09|
> > |MNLI(random)|0.8156|0.588|0.2276|9.02e-06|
> > |MNLI_MIS(pretrain)|0.9994|0.8453|0.1541|9.43e-10|
> > |MNLI_MIS(random)|0.8096|0.598|0.2116|5.28e-06|
> > |MRPC(pretrain)|0.9988|0.8211|0.1777|1.18e-04|
> > |MRPC(random)|0.9936|0.6593|0.3343|3.76e-03|
> >
> >
> > We observe that during fine-tuning, a lower IIW does not always correspond to the best test performance, which aligns with the conclusions of our paper. The following table shows the training performance, test performance, and generalization error of GPT-2 during the fine-tuning process on the RTE dataset. Notably, GPT-2 achieves its best test performance at epoch 2, but this does not correspond to the lowest IIW.
> >
> > |    RTE(GPT2)       | train_acc | test_acc | acc_gap | iiw      |
> > | ------------- | --------- | -------- | ------- | -------- |
> > |Epoch 0|0.4975|0.4729|0.02462|2.35e+01|
> > |Epoch 1|0.5533|0.556|0.0026|7.06e-02|
> > |Epoch 2|0.8282|0.6643|0.1639|6.63e-01|
> > |Epoch 3|0.9366|0.6426|0.294|3.94e-01|
> > |Epoch 4|0.9732|0.6606|0.3126|2.49e-02|
> > |Epoch 5|0.9964|0.6606|0.3358|2.08e-03|
> >
> > We also have added the experimental results of GPT-2 on the XSum and CNN/DailyMail summarization tasks. Perplexity (PPL) is used as the evaluation metric. We observe that pre-trained models achieve lower PPL while also exhibiting lower IIW.
> >
> > |    GPT2       | train_ppl | test_ppl | ppl_gap | iiw      |
> > | ------------- | --------- | -------- | ------- | -------- |
> > |xsum(pretrain)|2.321|8.055|5.734|2.10e-01|
> > |xsum(random)|59.98|127.9|67.95|1.08e+01|
> > |cnn_dailymail(pretrain)|2.023|6.623|4.6|1.61e-01|
> > |cnn_dailymail(random)|69.19|278.6|209.4|9.40e+00|
> >
> > The following are the results of GPT-2 during the fine-tuning process on the CNN/DailyMail dataset. It can be observed that the model achieves the smallest generalization error at Epoch 1, but the corresponding IIW is not the smallest. This is consistent with the conclusions of our previous experiments.
> >
> > |    cnn_dailymail       | train_ppl | test_ppl | ppl_gap | iiw      |
> > | ------------- | --------- | -------- | ------- | -------- |
> > |Epoch 0|13.53|11.21|2.32|1.31e+00|
> > |Epoch 1|4.002|5.503|1.501|1.74e-02|
> > |Epoch 2|3.264|5.564|2.3|5.07e-02|
> > |Epoch 3|2.729|5.834|3.105|1.05e-01|
> > |Epoch 4|2.313|6.271|3.958|1.41e-01|
> > |Epoch 5|2.023|6.623|4.6|1.61e-01|

---

> > > ### Comment · Reviewer_MWY2 · 2024-11-26
> > > **Acknowledgement**
> > >
> > > Thank you for the responses and for presenting additional experiment results. I definitely appreciate and acknowledge that the exploration of IIW in the context of language models fills an important gap in understanding the generalization capabilities of these models. However, there is no lear signal for correlation between IIW and model performance from the experimental results in my opinion. And for this paper to be conclusive, more explanation is needed for the inconsistency between IIW and test performance. I will keep my score as-is.

---

> > > > ### Author Response · Authors · 2024-11-27
> > > > **Rebuttal by Authors**
> > > >
> > > > We sincerely appreciate your response. To better understand the relationship between IIW and test performance, we provide the following statistical analysis.
> > > >
> > > > In our experiments (including GPT-2), pre-trained models consistently outperform non-pre-trained models and exhibit lower IIW. Thus, in such cases, IIW can reliably serve as a generalization proxy. We conducted a statistical analysis of the performance differences on the test set for pre-trained language models and non-pre-trained language models. As shown in the table below, when the performance difference exceeds 0.3478 or the IIW difference is greater than 4.14e-01, we can conclude that IIW can reliably serve as a generalization proxy.
> > > >
> > > > |Stats Info|Test Acc Gap| IIW Gap|
> > > > | ------------- | --------- | -------- |
> > > > |mean|0.2152|1.80e-02|
> > > > |median|0.218|6.13e-03|
> > > > |max|0.3478|4.14e-01|
> > > >
> > > > During the fine-tuning process of pre-trained models, we analyzed cases where the test performance improves but the IIW does not decrease. We calculated the IIW gap and performance gap in such situations. The analysis shows that when the performance difference is less than 0.1588 or the IIW gap is less than 5.09e-02, IIW cannot reliably serve as a generalization proxy. Considering the median, the IIW in these cases is already relatively small. The approximation error of IIW leads to an unstable range of 0.01716 in Test Accuracy.
> > > >
> > > > |Stats Info|Test Acc Gap| IIW Gap|
> > > > | ------------- | --------- | -------- |
> > > > |mean|0.03805|5.09e-04|
> > > > |median|0.01716|3.28e-06|
> > > > |max|0.1588|5.09e-02|
> > > >
> > > >
> > > > For all fine-tuning experiments with pre-trained models, we also compare using the lowest IIW as the model selection criterion against the best test performance. We find that the average difference between using IIW as the selection criterion and the best test performance is only 0.0053. This suggests that, although lower IIW does not always correspond to better test performance, using IIW as a model selection criterion is entirely acceptable in practical applications.
> > > >
> > > > |Stats Info|Test Acc Gap|
> > > > | ------------- | --------- |
> > > > |mean|0.0053|
> > > > |medium|0.0029|
> > > > |max|0.0220|

---

### Official Review · Reviewer_UgTu · 2024-11-04

**Soundness:** 3
**Presentation:** 3
**Contribution:** 3
**Rating:** 6
**Confidence:** 3

**Summary:**

The paper investigates whether Information in Weights (IIW), an information-theoretic metric, can effectively explain the generalization performance of pre-trained language models. The authors first explain the IIW metric from Xu&Raginsky et al.'17 and show that the metric is lower for pre-trained models compared to non pre-trained models. However, IIW does not reliably serve as a generalization proxy during the fine-tuning of pre-trained models. The authors study empirical and theoretical insights into precision concerns of the IIW metric, especially during fine-tuning, when the parameters are allowed to grow unbounded.

**Strengths:**

The strength of  the paper lies in its motivation to measure existing weight metrics to predict generalization guarantees for pre-trained language models. Through careful experiments, the authors show that the IIW metric can distinguish between pre-trained and non pre-trained models. However, the metric can't be used for predicting generalization guarantees after fine-tuning. Through a careful empirical and theoretical analysis, the authors dive into precision issues of the IIW metric.

In summary, the findings suggest that while IIW helps explain why pre-trained models outperform others, its utility is limited in fine-tuning and more precise methods are needed to serve as a reliable generalization proxy. Overall, this paper stands as an important step towards developing robust generalization metrics.

**Weaknesses:**

As such, I don't see clear weaknesses with the current work. I have a couple of questions regarding the experimental setup and the conclusion drawn from the observations.

a) **When is IIW computed?** In table 2, when comparing between FT and LP-FT, when is IIW computed for each training method? Is it computed at the end of training for each method? Furthermore, please consider the following questions:
- Will IIW at initialization be a measure that can different between the two methods? How would IIW for LP-FT after LP (first epoch) phase compare to IIW for FT at initialization?
- Can IIW present better correlation for regularized fine-tuning, where the weights are restricted to stay close to the pre-trained values? A plot comparing IIW, regularization strength, and generalization behavior can further strengthen the statement in proposition 1.
- Furthermore, Kumar et al.'22 proposed to use OOD generalization performance to differentiate between FT and LP-FT? Can IIW relate better to OOD generalization?

b) **Increasing IIW with layer depth:** Plots in figure 1 show that IIW of deeper layers are higher for a pre-training model. Is that a generic metric that could be used to select the optimal set of parameters to fine-tune for optimal/efficient training?

c) **Auto-regressive models like GPT-2:** How would the observations change for auto-regressive models like GPT? Specifically, how would the following observations change?
- IIW of the fully connected (fc) layer is significantly higher than that of other layers, because it's randomly initialized for BERT. However, this is not the case for GPT-2.
- Single token v/s multi token in table 1: How would the IIW behavior change with a fine-tuning task that involves multi-token generation at test time?

d) **Computation necessity for the fisher matrix:** Can the authors discuss about how expensive measuring the fisher matrix is (in terms of wall clock time and number of GPUs/CPUs necessary) for a 125M parameter model?
- How many training samples are necessary to approximate the fisher matrix well for the reported IIW scores?
- Also, are there approximate versions of fisher matrix (e.g. zeroth order approximations), that the authors could use to compute the IIW metric?

**Questions:**

Please check my questions in the section above.

---

> ### Author Response · Authors · 2024-11-24
> **Rebuttal by Authors**
>
> Thank you very much for your careful review. We appreciate your feedback and will do our best to answer your questions.
>
> ---
>
> Q: a) When is IIW computed? 1)In table 2, when comparing between FT and LP-FT, when is IIW computed for each training method? Is it computed at the end of training for each method? Furthermore, please consider the following questions:
> 2)Will IIW at initialization be a measure that can different between the two methods? How would IIW for LP-FT after LP (first epoch) phase compare to IIW for FT at initialization?
> 3)Can IIW present better correlation for regularized fine-tuning, where the weights are restricted to stay close to the pre-trained values? A plot comparing IIW, regularization strength, and generalization behavior can further strengthen the statement in proposition 1.
> 4)Furthermore, Kumar et al.'22 proposed to use OOD generalization performance to differentiate between FT and LP-FT? Can IIW relate better to OOD generalization?
>
> A: 1)During the experiments, we calculate IIW at each epoch. For all experiments, we consistently report the IIW and other performance metrics from the final epoch.
>
> 2)For a given dataset and model, the initial IIW is fixed. Therefore, for the same model, the initial stage IIW cannot be used to distinguish different training methods. The following are the performance metrics of BERT using LP-FT and FT on QQP. We can see that the first Epoch having a lower IIW does not necessarily result in the final Epoch having the best performance metrics.
>
> |    QQP(PT-FT)       | train_acc | test_acc | acc_gap | iiw      |
> | ------------- | --------- | -------- | ------- | -------- |
> |Epoch 0|0.3934|0.3947|0.001295|7.63e-07|
> |Epoch 1|0.7264|0.7291|0.002717|7.77e-07|
> |Epoch 5|0.9996|0.9103|0.0893|1.95e-09|
>
> |    QQP(FT)       | train_acc | test_acc | acc_gap | iiw      |
> | ------------- | --------- | -------- | ------- | -------- |
> |Epoch 0|0.3934|0.3947|0.001295|7.63e-07|
> |Epoch 1|0.9219|0.8909|0.03096|8.70e-08|
> |Epoch 5|0.9996|0.9094|0.0902|1.49e-09|
>
> 3)In fact, if the weights are forced to remain unchanged, the model will not be able to fit the training set well, even if the IIW at this time has a better approximation effect. Since IIW depends on weight gap and Fisher matrix, we will consider how to combine weight gap and Fisher matrix to design regularization methods in the future.
>
>
> 4)From the perspective of information-theoretic generalization bounds, the purpose of IIW is not related to OOD. We believe that the relationship between IIW and OOD can be explored theoretically first, and this is worth exploring as a direction for future research.
>
> ---

---

> > ### Author Response · Authors · 2024-11-24
> > **Rebuttal by Authors**
> >
> > Q: Increasing IIW with layer depth: Plots in figure 1 show that IIW of deeper layers are higher for a pre-training model. Is that a generic metric that could be used to select the optimal set of parameters to fine-tune for optimal/efficient training?
> >
> > A: We think this is a great suggestion. Due to the fact that the FC layer typically has the highest IIW, while the Embedding layer has a smaller IIW. Therefore, we only train the fc layer in the first epoch. The FC and middle layers are trained at Epochs 2-3, while all layers are trained at epochs 4-5. We call this method graded fine tuning (GFT). The following are the results of fine-tuning using GFT for BERT, RoBERTa, and ELECTRA. We can see that GFT performs better than FT on most datasets.
> >
> > |    Bert       | train_acc | test_acc | acc_gap | iiw      |
> > | ------------- | --------- | -------- | ------- | -------- |
> > |RTE(FT)|0.9991|0.6751|0.324|2.54e-03|
> > |RTE(GFT)|0.9982|0.6968|0.3015|9.07e-03|
> > |QNLI(FT)|0.9999|0.9107|0.08924|3.43e-09|
> > |QNLI(GFT)|0.9997|0.9129|0.08686|4.22e-08|
> > |QQP(FT)|0.9996|0.9094|0.0902|1.49e-09|
> > |QQP(GFT)|0.9995|0.9103|0.08928|6.78e-10|
> > |SST2(FT)|0.9984|0.9197|0.07872|9.60e-09|
> > |SST2(GFT)|0.9984|0.922|0.07638|7.33e-08|
> > |MNLI(FT)|0.9992|0.8358|0.1634|7.10e-09|
> > |MNLI(GFT)|0.999|0.8398|0.1592|2.10e-09|
> > |MNLI_MIS(FT)|0.9992|0.8361|0.163|3.53e-09|
> > |MNLI_MIS(GFT)|0.999|0.8406|0.1584|2.12e-09|
> > |MRPC(FT)|0.9997|0.8162|0.1835|1.50e-04|
> > |MRPC(GFT)|0.9994|0.8456|0.1538|4.03e-04|
> >
> > |    Electra    | train_acc | test_acc | acc_gap | iiw      |
> > | ------------- | --------- | -------- | ------- | -------- |
> > |RTE(FT)|0.9991|0.787|0.2121|2.03e-03|
> > |RTE(GFT)|0.9991|0.7942|0.2049|4.32e-03|
> > |QNLI(FT)|0.9998|0.9218|0.07795|3.21e-08|
> > |QNLI(GFT)|0.9998|0.9242|0.07555|2.28e-08|
> > |QQP(FT)|0.9994|0.9204|0.07905|4.61e-09|
> > |QQP(GFT)|0.9992|0.92|0.07919|5.83e-09|
> > |SST2(FT)|0.9981|0.9392|0.05883|7.37e-08|
> > |SST2(GFT)|0.9975|0.9518|0.04562|1.01e-07|
> > |MNLI(FT)|0.999|0.8744|0.1247|4.65e-09|
> > |MNLI(GFT)|0.9988|0.8819|0.1169|1.11e-08|
> > |MNLI_MIS(FT)|0.999|0.8764|0.1226|4.65e-09|
> > |MNLI_MIS(GFT)|0.9988|0.8843|0.1146|1.11e-08|
> > |MRPC(FT)|0.9985|0.8873|0.1112|5.31e-04|
> > |MRPC(GFT)|0.9988|0.8897|0.1091|1.22e-03|
> >
> > |   Roberta    | train_acc | test_acc | acc_gap | iiw      |
> > | ------------- | --------- | -------- | ------- | -------- |
> > |RTE(FT)|0.992|0.7581|0.2338|5.37e-04|
> > |RTE(GFT)|0.9893|0.787|0.2023|3.37e-04|
> > |QNLI(FT)|0.9995|0.9196|0.0799|1.48e-08|
> > |QNLI(GFT)|0.9992|0.9226|0.07664|6.57e-08|
> > |QQP(FT)|0.9986|0.9107|0.08793|2.11e-09|
> > |QQP(GFT)|0.9973|0.9126|0.08466|1.31e-08|
> > |SST2(FT)|0.9974|0.9346|0.06274|6.92e-08|
> > |SST2(GFT)|0.9971|0.9381|0.05901|1.33e-07|
> > |MNLI(FT)|0.9981|0.862|0.1361|4.00e-09|
> > |MNLI(GFT)|0.9973|0.8663|0.1309|7.97e-09|
> > |MNLI_MIS(FT)|0.9981|0.8621|0.136|4.00e-09|
> > |MNLI_MIS(GFT)|0.9973|0.8653|0.1319|7.97e-09|
> > |MRPC(FT)|0.9979|0.8922|0.1057|1.53e-04|
> > |MRPC(GFT)|0.997|0.8995|0.09746|7.61e-05|

---

> > > ### Author Response · Authors · 2024-11-24
> > > **Rebuttal by Authors**
> > >
> > > Q: c) Auto-regressive models like GPT-2: 1)How would the observations change for auto-regressive models like GPT? Specifically, how would the following observations change?
> > > 2)IIW of the fully connected (fc) layer is significantly higher than that of other layers, because it's randomly initialized for BERT. However, this is not the case for GPT-2.
> > > 3)Single token v/s multi token in table 1: How would the IIW behavior change with a fine-tuning task that involves multi-token generation at test time?
> > >
> > > A: 1）3）We have added the experimental results for GPT-2 (Medium). The training performance, test performance, generalization error, and IIW of the pre-trained GPT-2 model and non-pre-trained model are shown in the table below. It is evident that pre-trained models consistently exhibit lower IIW while achieving better test performance.
> > >
> > > |    GPT2       | train_acc | test_acc | acc_gap | iiw      |
> > > | ------------- | --------- | -------- | ------- | -------- |
> > > |RTE(pretrain)|0.9964|0.6606|0.3358|2.08e-03|
> > > |RTE(random)|0.9652|0.4693|0.4959|1.55e-02|
> > > |QNLI(pretrain)|0.9998|0.9109|0.08898|3.96e-09|
> > > |QNLI(random)|0.9617|0.5795|0.3822|2.17e-05|
> > > |QQP(pretrain)|0.9997|0.9075|0.09212|3.44e-10|
> > > |QQP(random)|0.9787|0.7986|0.1801|1.04e-06|
> > > |SST2(pretrain)|0.9985|0.9358|0.0627|3.36e-08|
> > > |SST2(random)|0.983|0.7982|0.1848|3.94e-06|
> > > |MNLI(pretrain)|0.9994|0.8446|0.1548|1.01e-09|
> > > |MNLI(random)|0.8156|0.588|0.2276|9.02e-06|
> > > |MNLI_MIS(pretrain)|0.9994|0.8453|0.1541|9.43e-10|
> > > |MNLI_MIS(random)|0.8096|0.598|0.2116|5.28e-06|
> > > |MRPC(pretrain)|0.9988|0.8211|0.1777|1.18e-04|
> > > |MRPC(random)|0.9936|0.6593|0.3343|3.76e-03|
> > >
> > >
> > > We observe that during fine-tuning, a lower IIW does not always correspond to the best test performance, which aligns with the conclusions of our paper. The following table shows the training performance, test performance, and generalization error of GPT-2 during the fine-tuning process on the RTE dataset. Notably, GPT-2 achieves its best test performance at epoch 2, but this does not correspond to the lowest IIW.
> > >
> > > |    RTE(GPT2)       | train_acc | test_acc | acc_gap | iiw      |
> > > | ------------- | --------- | -------- | ------- | -------- |
> > > |Epoch 0|0.4975|0.4729|0.02462|2.35e+01|
> > > |Epoch 1|0.5533|0.556|0.0026|7.06e-02|
> > > |Epoch 2|0.8282|0.6643|0.1639|6.63e-01|
> > > |Epoch 3|0.9366|0.6426|0.294|3.94e-01|
> > > |Epoch 4|0.9732|0.6606|0.3126|2.49e-02|
> > > |Epoch 5|0.9964|0.6606|0.3358|2.08e-03|
> > >
> > > We also have added the experimental results of GPT-2 on the XSum and CNN/DailyMail summarization tasks. Perplexity (PPL) is used as the evaluation metric. We observe that pre-trained models achieve lower PPL while also exhibiting lower IIW.
> > >
> > > |    GPT2       | train_ppl | test_ppl | ppl_gap | iiw      |
> > > | ------------- | --------- | -------- | ------- | -------- |
> > > |xsum(pretrain)|2.321|8.055|5.734|2.10e-01|
> > > |xsum(random)|59.98|127.9|67.95|1.08e+01|
> > > |cnn_dailymail(pretrain)|2.023|6.623|4.6|1.61e-01|
> > > |cnn_dailymail(random)|69.19|278.6|209.4|9.40e+00|
> > >
> > > The following are the results of GPT-2 during the fine-tuning process on the CNN/DailyMail dataset. It can be observed that the model achieves the smallest generalization error at Epoch 1, but the corresponding IIW is not the smallest. This is consistent with the conclusions of our previous experiments.
> > >
> > > |    cnn_dailymail       | train_ppl | test_ppl | ppl_gap | iiw      |
> > > | ------------- | --------- | -------- | ------- | -------- |
> > > |Epoch 0|13.53|11.21|2.32|1.31e+00|
> > > |Epoch 1|4.002|5.503|1.501|1.74e-02|
> > > |Epoch 2|3.264|5.564|2.3|5.07e-02|
> > > |Epoch 3|2.729|5.834|3.105|1.05e-01|
> > > |Epoch 4|2.313|6.271|3.958|1.41e-01|
> > > |Epoch 5|2.023|6.623|4.6|1.61e-01|
> > >
> > > 2）We compared the IIW between the Embedding layer and the final layer of randomly initialized and pre trained GPT2 on the XSum and CNN/DailyMail summarization datasets. Similar to models like BERT, the weight of the randomly initialized Embedding layer is still very large. The difference lies in that, since pre-trained GPT2 does not require random initialization of the last layer, its IIW of FC is much smaller than that of the Embedding layer.
> > >
> > > |    GPT2           |    Layer()      | IIW |
> > > | ------------- | ------------- | --------- |
> > > |XSum(pretrain)| Embedding | 42.69 |
> > > |XSum(pretrain)| Last Layer | 0.00026|
> > > |XSum(random)| Embedding | 3070.16 |
> > > |XSum(random)| Last Layer | 8.76 |
> > > |CNN/DailyMail(pretrain)| Embedding | 2692.06 |
> > > |CNN/DailyMail(pretrain)| Last Layer | 0.623 |
> > > |CNN/DailyMail(random)| Embedding | 33.01 |
> > > |CNN/DailyMail(random)| Last Layer | 4.89|

---

> > > > ### Author Response · Authors · 2024-11-24
> > > > **Rebuttal by Authors**
> > > >
> > > > Q: d) Computation necessity for the fisher matrix: 1)Can the authors discuss about how expensive measuring the fisher matrix is (in terms of wall clock time and number of GPUs/CPUs necessary) for a 125M parameter model?
> > > > 2)How many training samples are necessary to approximate the fisher matrix well for the reported IIW scores?
> > > > 3)Also, are there approximate versions of fisher matrix (e.g. zeroth order approximations), that the authors could use to compute the IIW metric?
> > > >
> > > > A: 1）For BERT, we performed calculations on two GTX 4090Ti graphics cards. It takes about 5 minutes to calculate IIW once.
> > > >
> > > > 2) In our experiment, we used a total of 5k training samples for a single IIW calculation.
> > > >
> > > > 3）As far as we know, there is currently no other method available for calculating Fisher Matrix.

---

> ### Comment · Reviewer_UgTu · 2024-12-02
>
> I thank the authors for their detailed responses. I believe that this is a good paper. However, I keep my score as-is as I agree with reviewer aaJ2's comments. In the next version, please include discussions on how IIW can be useful to predict generalization trends. GFT is an important contribution, please include the basis of design choices behind SFT as well.

---

### Official Review · Reviewer_v2DU · 2024-11-04

**Soundness:** 1
**Presentation:** 2
**Contribution:** 2
**Rating:** 3
**Confidence:** 3

**Summary:**

A host of results demonstrate that information stored in weights (IIW) provide an upper bound on generalization error. In this work, the authors demonstrate that IIW can be used to explain why pretrained language models generalize better compared to non-pretrained language models. However, they also demonstrate that this metric is not sufficient to disginuish the generalizability between finetuned language models.

**Strengths:**

- The paper seeks to understand the factors that drive the performance of pretrained language. Given the widespread adoption of language models, this is a timely topic.
- Moreover, the paper addresses several empirical issues with IIW which I think is important for practitioners especially those working in uncertainty quantification.

**Weaknesses:**

Overall, I feel that many claims in the paper are not well-supported and the scope of the experiments are quite limited.

- It is hard for me to pin down exactly what the contributions of the paper are. For example, I could see the paper trying to answer any one of the following questions:
    - What drives the performance of pre-trained language models compared to non-pretrained language models?
    - Can IIW generalization bounds characterize the inductive biases of classes of language models or our learning algorithms?
    - How tight are IIW generalization bounds on language models?

    It seems that the paper is trying to do all three, but each one is done at a superficial level.

- All of the experiments in the paper focus on encoder-only language models. This makes the scope of the paper quite limited. It makes me question whether or not these results would generalize to decoder-only language models.
- The tasks being explored are only binary (or ternary) classification tasks. What about more complex tasks of interest to the NLP community like summarization or generative benchmarks?
- Much of the analysis being performed is purely qualitative. Even claims that could be made quantitative or statistically analyzed at not (see lines 292-293, lines 284-286).
- The tables in the paper are hard to interpret (Table 1, Table 2). It is unclear what conclusions should be drawn from these results.
- In lines 203-206, lines 210-213 bold claims are being made without any supporting evidence. For example, why does higher IIW correspond to “memorization,” this relationship is nontrivial. Or what does “knowledge acquired” mean here?

**Questions:**

- The tables in the paper, illustrate the relationship between IIW and test(?) accuracy. However, the theoretical bound presented is about generalization gap. Are the authors assuming that the training accuracy is always some constant like 100%? If this is not the case, then I doubt the validity of the subsequent analyses.
- The plots in the paper (Fig. 1, Fig. 2) measure IIW at a low granularity–every epoch. Why did the authors choose to do this? Could the rising/falling trajectories of IIW discussed in the literature be more apparent if the authors instead probed IIW every $n$ steps?

---

> ### Author Response · Authors · 2024-11-24
> **Rebuttal by Authors**
>
> Thank you very much for your careful review. We appreciate your feedback and will do our best to answer your questions.
>
> ---
>
> Q1: It is hard for me to pin down exactly what the contributions of the paper are. For example, I could see the paper trying to answer any one of the following questions:...
>
> A: In this paper, we primarily investigate whether information-theoretic generalization bounds can be used to explain the effectiveness of pre-trained language models. Specifically, we explore whether these bounds can directly measure the generalization performance of pre-trained language models. To validate this, we experimentally compare the IIW (Information in Weights) of pre-trained and non-pre-trained language models and find that pre-trained models exhibit tighter generalization bounds. Since IIW depends solely on the training dataset, we further examine whether IIW can be directly used during the fine-tuning process as a criterion for model selection without relying on the test set. However, experimental results show that models with lower IIW do not always exhibit better performance, creating an inconsistency with the information-theoretic generalization bounds. We analyze this phenomenon and demonstrate, both theoretically and empirically, that current methods for estimating IIW involve certain inaccuracies, which may explain why IIW cannot consistently serve as a reliable proxy for model generalization.
>
> ---
>
> Q2: All of the experiments in the paper focus on encoder-only language models. This makes the scope of the paper quite limited. It makes me question whether or not these results would generalize to decoder-only language models.
>
> A: We have added the experimental results for GPT-2 (Medium). The training performance, test performance, generalization error, and IIW of the pre-trained GPT-2 model and non-pre-trained model are shown in the table below. It is evident that pre-trained models consistently exhibit lower IIW while achieving better test performance.
>
> |    GPT2       | train_acc | test_acc | acc_gap | iiw      |
> | ------------- | --------- | -------- | ------- | -------- |
> |RTE(pretrain)|0.9964|0.6606|0.3358|2.08e-03|
> |RTE(random)|0.9652|0.4693|0.4959|1.55e-02|
> |QNLI(pretrain)|0.9998|0.9109|0.08898|3.96e-09|
> |QNLI(random)|0.9617|0.5795|0.3822|2.17e-05|
> |QQP(pretrain)|0.9997|0.9075|0.09212|3.44e-10|
> |QQP(random)|0.9787|0.7986|0.1801|1.04e-06|
> |SST2(pretrain)|0.9985|0.9358|0.0627|3.36e-08|
> |SST2(random)|0.983|0.7982|0.1848|3.94e-06|
> |MNLI(pretrain)|0.9994|0.8446|0.1548|1.01e-09|
> |MNLI(random)|0.8156|0.588|0.2276|9.02e-06|
> |MNLI_MIS(pretrain)|0.9994|0.8453|0.1541|9.43e-10|
> |MNLI_MIS(random)|0.8096|0.598|0.2116|5.28e-06|
> |MRPC(pretrain)|0.9988|0.8211|0.1777|1.18e-04|
> |MRPC(random)|0.9936|0.6593|0.3343|3.76e-03|
>
>
> We observe that during fine-tuning, a lower IIW does not always correspond to the best test performance, which aligns with the conclusions of our paper. The following table shows the training performance, test performance, and generalization error of GPT-2 during the fine-tuning process on the RTE dataset. Notably, GPT-2 achieves its best test performance at epoch 2, but this does not correspond to the lowest IIW.
>
> |    RTE(GPT2)       | train_acc | test_acc | acc_gap | iiw      |
> | ------------- | --------- | -------- | ------- | -------- |
> |Epoch 0|0.4975|0.4729|0.02462|2.35e+01|
> |Epoch 1|0.5533|0.556|0.0026|7.06e-02|
> |Epoch 2|0.8282|0.6643|0.1639|6.63e-01|
> |Epoch 3|0.9366|0.6426|0.294|3.94e-01|
> |Epoch 4|0.9732|0.6606|0.3126|2.49e-02|
> |Epoch 5|0.9964|0.6606|0.3358|2.08e-03|
>
> ---
>
> Q3: The tasks being explored are only binary (or ternary) classification tasks. What about more complex tasks of interest to the NLP community like summarization or generative benchmarks?
>
> A: We have added the experimental results of GPT-2 on the XSum and CNN/DailyMail summarization tasks. Perplexity (PPL) is used as the evaluation metric. We observe that pre-trained models achieve lower PPL while also exhibiting lower IIW.
>
> |    GPT2       | train_ppl | test_ppl | ppl_gap | iiw      |
> | ------------- | --------- | -------- | ------- | -------- |
> |xsum(pretrain)|2.321|8.055|5.734|2.10e-01|
> |xsum(random)|59.98|127.9|67.95|1.08e+01|
> |cnn_dailymail(pretrain)|2.023|6.623|4.6|1.61e-01|
> |cnn_dailymail(random)|69.19|278.6|209.4|9.40e+00|
>
> The following are the results of GPT-2 during the fine-tuning process on the CNN/DailyMail dataset. It can be observed that the model achieves the smallest generalization error at Epoch 1, but the corresponding IIW is not the smallest. This is consistent with the conclusions of our previous experiments.
>
> |    cnn_dailymail       | train_ppl | test_ppl | ppl_gap | iiw      |
> | ------------- | --------- | -------- | ------- | -------- |
> |Epoch 0|13.53|11.21|2.32|1.31e+00|
> |Epoch 1|4.002|5.503|1.501|1.74e-02|
> |Epoch 2|3.264|5.564|2.3|5.07e-02|
> |Epoch 3|2.729|5.834|3.105|1.05e-01|
> |Epoch 4|2.313|6.271|3.958|1.41e-01|
> |Epoch 5|2.023|6.623|4.6|1.61e-01|

---

> > ### Author Response · Authors · 2024-11-24
> > **Rebuttal by Authors**
> >
> > Q4: The tables in the paper are hard to interpret (Table 1, Table 2). It is unclear what conclusions should be drawn from these results.
> >
> > A: Table 1 demonstrates that pre-trained models have lower IIW, which corresponds to a tighter upper bound on the generalization error, leading to better performance. Table 2 highlights that IIW cannot consistently serve as a reliable proxy for generalization error. For example, the results of fine-tuning BERT on the MRPC dataset using two methods, FT and LP-FT, are shown below. The BERT model fine-tuned with LP-FT achieves better performance but exhibits a higher IIW.
> >
> > |    MRPC(BERT)       | train_acc | test_acc | acc_gap | iiw      |
> > | ------------- | --------- | -------- | ------- | -------- |
> > | FT  |0.9997|0.8162|0.1835|1.50e-04|
> > | LP-FT |0.9994|0.8456|0.1538|4.03e-04|
> >
> > ---
> >
> > Q5: In lines 203-206, lines 210-213 bold claims are being made without any supporting evidence. For example, why does higher IIW correspond to “memorization,” this relationship is nontrivial. Or what does “knowledge acquired” mean here?
> >
> > A: IIW, represented as $I(W,Z)$, denotes the mutual information between the model WW and the training dataset $Z$. If $W$ contains more information about $Z$, the IIW value will be larger. When $I(W,Z)=H(Z)$,  $I(W,Z)$ fully captures the information of $Z$, which implies that WW completely memorizes $Z$. However, a larger IIW corresponds to a higher generalization error.
> >
> > From the perspective of information-theoretic generalization bounds, we aim for a model that fits the training data well while maintaining a lower IIW. Experimental results indicate that compared to randomly initialized models, pre-trained models contain less information from the training dataset. This suggests that during pre-training, the model learns some information (or "knowledge") that reduces its dependency on the training dataset during downstream fine-tuning.
> >
> > ---
> >
> > Q6: The tables in the paper, illustrate the relationship between IIW and test(?) accuracy. However, the theoretical bound presented is about generalization gap. Are the authors assuming that the training accuracy is always some constant like 100%? If this is not the case, then I doubt the validity of the subsequent analyses.
> >
> > A: We do not assume that training accuracy is always a constant, such as 100%. From the GPT2 experimental results in Q2, we observe that models achieving better test performance after training typically also exhibit lower generalization error.
> >
> > ---
> >
> > Q7: The plots in the paper (Fig. 1, Fig. 2) measure IIW at a low granularity–every epoch. Why did the authors choose to do this? Could the rising/falling trajectories of IIW discussed in the literature be more apparent if the authors instead probed IIW every n  steps?
> >
> > A: PAC-IB [1] calculates IIW for VGG16-Net at each epoch during its comparative experiments, and we follow the same setup here. In practice, the interval step for computing IIW does not affect the rising/falling trajectories of IIW trends.
> >
> > [1]PAC-Bayes Information Bottleneck

---

> ### Comment · Reviewer_v2DU · 2024-11-27
>
> Thank you for the detailed response and additional experiments. I understand the scope and intended contribution of the paper much better now. However, it is still difficult for me to see the causal relationship (or even sufficiency) relationship between IIW and generalizability, in practice (I understand that it leads to tighter bounds on generalizability, but loose bounds do not necessarily imply good generalization; and it seems that the authors' own summary of their work agree with this statement). As a result, I keep my score as is.

---

> > ### Author Response · Authors · 2024-11-27
> > **Rebuttal by Authors**
> >
> > We sincerely appreciate your response. To better understand the relationship between IIW and test performance, we provide the following statistical analysis.
> >
> > In our experiments (including GPT-2), pre-trained models consistently outperform non-pre-trained models and exhibit lower IIW. Thus, in such cases, IIW can reliably serve as a generalization proxy. We conducted a statistical analysis of the performance differences on the test set for pre-trained language models and non-pre-trained language models. As shown in the table below, when the performance difference exceeds 0.3478 or the IIW difference is greater than 4.14e-01, we can conclude that IIW can reliably serve as a generalization proxy.
> >
> > |Stats Info|Test Acc Gap| IIW Gap|
> > | ------------- | --------- | -------- |
> > |mean|0.2152|1.80e-02|
> > |median|0.218|6.13e-03|
> > |max|0.3478|4.14e-01|
> >
> > During the fine-tuning process of pre-trained models, we analyzed cases where the test performance improves but the IIW does not decrease. We calculated the IIW gap and performance gap in such situations. The analysis shows that when the performance difference is less than 0.1588 or the IIW gap is less than 5.09e-02, IIW cannot reliably serve as a generalization proxy. Considering the median, the IIW in these cases is already relatively small. The approximation error of IIW leads to an unstable range of 0.01716 in Test Accuracy.
> >
> > |Stats Info|Test Acc Gap| IIW Gap|
> > | ------------- | --------- | -------- |
> > |mean|0.03805|5.09e-04|
> > |median|0.01716|3.28e-06|
> > |max|0.1588|5.09e-02|
> >
> >
> > For all fine-tuning experiments with pre-trained models, we also compare using the lowest IIW as the model selection criterion against the best test performance. We find that the average difference between using IIW as the selection criterion and the best test performance is only 0.0053. This suggests that, although lower IIW does not always correspond to better test performance, using IIW as a model selection criterion is entirely acceptable in practical applications.
> >
> > |Stats Info|Test Acc Gap|
> > | ------------- | --------- |
> > |mean|0.0053|
> > |medium|0.0029|
> > |max|0.0220|

---

### Note · Authors · 2024-12-04

I have read and agree with the venue's withdrawal policy on behalf of myself and my co-authors.